# Economic Reform, Labour Markets and Informal Sector Employment: Evidence from India

## Nihar Shembavnekar

Department of Economics, University of Sussex, Brighton BN1 9RH, UK; N.Shembavnekar@sussex.ac.uk

**Abstract:** Theory and economic intuition suggest that domestic institutions influence the employment impact of economic reform, but the evidence base is thin. This paper seeks to address this by examining the extent to which differences in regional labour market flexibility shaped the impact of unanticipated economic reforms on employment in informal (unregistered) manufacturing enterprises in India (1990–2001). It employs a difference-in-differences strategy and finds that tariff reductions are not associated with significant employment shifts in informal enterprises, a finding that may be attributable to the fact that these enterprises rarely engage in international trade. However, on average and ceteris paribus, delicensing (FDI reform) is associated with statistically significant increases (increases) in informal employment and informal enterprise numbers in inflexible (flexible) labour markets. There is some evidence that the delicensing effect is attributable to increases in product market competition in delicensed industries. However, the channel underlying the result associated with FDI reform is less clear. In light of the persistent primacy of the informal sector in India and other developing economies, these findings have substantial policy relevance.

**Keywords:** economic reform; informal sector; employment; labour market flexibility; India

## 1. Introduction

In the latter half of the twentieth century, several developing economies initiated comprehensive economic reform policies. A balance-of-payments crisis necessitating IMF assistance, preceded by a period of tepid growth and a growing realisation that the status quo was unsustainable, was the trigger for economic liberalisation in India in 1991. The Indian government implemented a series of far reaching economic reforms in the 1991–1997 period. The labour market impacts of these reforms remain somewhat poorly studied. Several studies, Nunn and Trefler (2013) and Ahsan (2013) being among the more recent, have documented that this impact is likely to be influenced by domestic institutions. However, this view has received scant attention in the Indian context, in particular at a 'micro' or firm level. Existing research also largely avoids the issue of economic duality that is typical of India and other developing economies. Put differently, the literature rarely distinguishes between registered, or formal, manufacturing firms and unregistered, or informal, manufacturers which, under Indian regulation, are manufacturing enterprises employing fewer than ten (twenty) workers and using (not using) electricity.

Estimated to account for 99 percent of businesses and approximately 80 percent of employment in the Indian manufacturing sector (Ghani et al. 2013a), economic outcomes in the informal sector merit as much academic and policy interest as those in the formal sector. More generally, the informal sector has long been an area of interest for development economists, since Hart's (1973) seminal study of informal enterprises. Charmes (1998) documented that even as late as in 1998, informal enterprises accounted for a substantial majority of overall employment in many developing economies in Asia,

sub-Saharan Africa and Latin America. A non-negligible proportion of academic work on the informal sector deals with conceptual and measurement issues (see for instance Feige 1990; Charmes 1998; Maloney 2004; and Henley et al. 2009).The parameters employed by varying paradigms are numerous and include the type of work undertaken, productivity, geography, formal-sector linkages and the degree of marginality, and are well summed up in Blunch et al. (2001). The policy implications of economic reform for the informal sector remain rather poorly understood, principally because good data on informal enterprises are rare and tend to be difficult to marry with reform episodes.

In that context, this paper is unique and timely. It seeks to address a glaring gap in the academic literature by analysing the impact of India's economic reforms in the 1990s on employment in small, informal manufacturing enterprises. I also examine the extent to which this impact depends on differences in labour market flexibility at the state (provincial) level. This is key, given that inflexible labour market regulation is commonly cited as an impediment to investment and growth in manufacturing output and productivity (Ahsan and Pagés 2009). I capture state-level variations in labour market flexibility using the 'FLEX 2' indicator proposed by Hasan et al. (2012). Unless otherwise specified, I use the terms 'states with flexible labour markets' and 'states with inflexible labour markets' to refer to states that are characterised as having flexible labour markets (score 1) and inflexible labour markets (score 0) by this 'FLEX 2' variable. Described in detail in Section 3, this indicator builds on the seminal state-level labour legislation-based measure proposed by Besley and Burgess (2004) by accounting for perceptions regarding the effectiveness of implementation of legislation.

While the wider Indian economy underwent substantial structural transformation in the 1990–2010 period, informal enterprises have stubbornly continued to account for a large proportion of India's manufacturing sector (Ghani et al. 2013a). Despite the wide-ranging economic reforms undertaken in the 1990s, the informal sector continues to corner significant chunks of employment in Indian manufacturing, across states and major industry groups, which means that understanding the drivers of informality necessitates an enterprise-level or state-industry-level analysis (Ghani et al. 2013a). As discussed in Section 3, the data reveal that a great majority of informal manufacturers are tiny household enterprises ('own account manufacturing enterprises', or OAMEs). Obtaining insights into whether economic reforms have any implications for employment in this informal manufacturing sector is of great policymaking interest, particularly in light of the findings of Kathuria et al. (2013), which show that market liberalisation is likely to widen the productivity gulf between formal and informal manufacturing businesses. Furthermore, the informal sector registers lower levels of education, income and value added relative to formal firms, due to which policymakers are also interested in incentivising worker flows from the informal sector into the formal sector (Unni 2005).

Informal enterprises rarely engage directly in international trade (this is revealed in governmental surveys of the informal sector), were not targeted by India's licensing regime and FDI caps, and are not subject to the labour market regulations with which formal firms are legally bound to comply. Any interactions between economic reform and states' labour market flexibility are therefore arguably unlikely to have a direct effect on employment in the informal sector. However, there is some evidence that there are linkages between the formal and informal manufacturing sectors in India. On the one hand, these linkages may be rooted in factors such as vertical integration, outsourcing and agglomeration, as suggested by Mukim (2013) and Ghani et al. (2013a, 2013b). On the other, as posited in recent work by Allen and Allen and Schipper (2017), formal and informal manufacturers may compete in some industries. The conclusions of Kathuria et al. (2013) are supportive of the hypothesis that India's reform programme affected enterprise-level efficiency in the informal manufacturing sector, with net productivity gains accruing to informal manufacturers (albeit, on average, to a lower degree relative to formal firms). Accounting for spillover effects in the informal sector arising from the interaction of economic reform and labour market flexibility is therefore of relevance for policy makers. I examine the plausibility of alternative channels, in particular that of product market competition, in this paper, although the fact that my dataset is not a panel precludes a rigorous analysis of market entry and exit.

India offers exceptionally fertile territory for a study of the impact of economic liberalization on informal sector employment, with regular governmental surveys of informal enterprises being complemented by a wide-ranging and sudden episode of economic reform in the 1990s. The analysis in this paper takes advantage of these national oddities by harnessing the informal sector survey data, compiled by the Indian National Sample Survey Office (NSSO) through quinquennial surveys of informal manufacturing enterprises. It also benefits from the rich cross-industry variation in India's policy changes in the 1990s, which is particularly visible in the import tariff reductions that were enforced. The reform package of 1991 was an unanticipated event, which helps to obviate the usual concerns inherent in any analysis of the consequences of such measures. This paper is the first to examine the impact of declines in both final goods and input tariffs on employment in informal enterprises in India. Its findings contribute to the literature by establishing that India's delicensing and FDI reforms are associated with significant shifts in informal sector employment.

The results indicate that reductions in final goods and input tariffs are not associated with significant employment shifts at the informal enterprise level. This is not counter-intuitive, given that the survey data show that informal enterprises rarely engage directly in international trade and global value chains. However, on average and ceteris paribus, delicensing is associated with a statistically significant increase of 10.8 percent in informal enterprise-level employment in states with inflexible labour markets, while no corresponding significant change is registered in states with flexible labour markets. FDI reform, on the other hand, is associated with informal enterprise-level employment rising by an average of 9.9 percent only in states with flexible labour markets, ceteris paribus, with no corresponding significant change visible in states with inflexible labour markets. The fact that neither the delicensing nor the FDI reforms directly targeted informal enterprises renders these findings particularly interesting, suggesting as it does that the informal sector may have experienced spillover impacts through linkages with larger formal firms, which bore the brunt of the direct effect of the reforms.

At a broader, three-digit industry level, delicensing goes hand-in-hand with informal enterprise numbers rising by 32 percent in states with inflexible labour markets, while FDI reform is associated with a corresponding increase of over 51 percent in states with flexible labour markets. These increases are restricted to tiny, household-only informal manufacturers, as opposed to slightly larger enterprises that hire outside labour. In line with the enterprise-level findings, no significance attaches to final goods reductions and input tariff declines.

There is some evidence that the delicensing effect is attributable to increases in product market competition between formal and informal entities. Greater competition within the formal sector also appears to be a predictor of informal sector expansion in recently delicensed industries, possibly because structural shifts of the type predicted in Melitz (2003). The mechanism underlying the result associated with FDI liberalisation is more uncertain, and could be one or a combination of competition or collaborative linkages between informal and formal manufacturers.

The remainder of this paper is organised as follows. Section 2 reviews the literature and provides context. Section 3 describes the data and Section 4 discusses the empirical methodology. Section 5 presents the main findings and several robustness checks. Section 6 concludes.

## 2. Context

The turn of the millennium witnessed an upsurge in academic interest in the impacts of tariff liberalisation programmes on firm-level employment, both in terms of theoretical contributions and empirical work. The literature has largely focused on final goods tariff declines, with substantial ambiguity persisting as regards employment effects. When it comes to distinguishing between the formal and informal sectors, informality has commonly been modelled at the individual or employee level. This may be attributable to the fact that a majority of studies exploit micro data from Latin American economies, most prominently from Brazil, that permit the identification of worker-level informality (see for instance Goldberg and Pavcnik 2003; Soares 2005; Aleman-Castilla 2006;

Bosch et al. 2007; Fugazza and Fiess 2010; and Paz 2014). In the Indian context, given that informality is captured at the enterprise level rather than the worker level, the relevance of these studies is limited.

As discussed in Section 1, economic liberalisation is perhaps unlikely to have directly affected employment in informal manufacturing enterprises in India. Nonetheless, to the extent that there are linkages between the formal and informal sectors, either on account of product market competition or through 'collaborative' supply chains, it is easy to imagine how a policy change that affects employment in formal firms might lead to changes in informal sector employment. For instance, if final goods tariff cuts and delicensing engender increased product market competition among formal firms in a given industry, the least productive formal firms might exit the market over time, through a Melitz (2003)-type market restructuring process. This could arguably be followed by a rise in informality in that industry, as the informal sector picks up some of the 'slack' in the labour market. This story could be expected to be stronger in industries that are also characterised by a greater degree of product market competition between formal and informal (as opposed to only among formal) operators, where the informal sector is more likely to function as a 'shock absorber' for a net employment loss in the formal sector.

On the other hand, if the formal firms and informal enterprises in an industry are engaged in collaborative linkages, it could be argued that an observed increase in formal sector employment would be complemented by a larger informal sector in that industry, and vice versa. In this sense, although the direct impact of input tariff declines and FDI reform might be restricted to formal firms, there could be spillovers into employment in informal enterprises. Although input tariff declines, for example, apply to imported inputs in the first instance, they could trigger general equilibrium effects in an industry's input supplying sectors and, over time, impose downward pressure on domestic input prices. In such a scenario, informal enterprise employment could respond to the input tariff reductions even if, as seems realistic, informal enterprises do not use sophisticated imported inputs and only source inputs locally. Along similar lines, if informal enterprises comprise a part of the supply chain of formal firms that are recipients of FDI inflows, informal sector employment may respond to FDI regime reform even while FDI flows only into a subset of formal firms. To the extent that individual policies might drive positive and negative changes in informal employment, which might in part cancel each other out, any impacts of significance that I observe in this study constitute net effects. I also attempt to disentangle the mechanisms underlying observed effects.

Furthermore, any such policy spillovers into informality might also be expected to differ across states with more and less flexible labour markets. Although informal enterprises themselves are not subject to the state-level labour market regulations outlined in Section 3, variations in the flexibility of state-level labour market regulation could influence any indirect impact that the reforms might have on these enterprises. It has been established, for instance, that Indian states with less flexible labour markets were characterised by higher levels of informal sector output even in 1992 (Besley and Burgess 2004).

The literature has implicitly tended to assume that formal and informal enterprises compete to gain market share. However, as Munro (2011) documents, a scenario in which the formal and informal sectors complement each other may constitute a more realistic description of developing economies. Complementarities could exist between and within the formal and informal sectors and might arise, for instance, through supply chain linkages or agglomeration driven externalities. As such, forward and backward linkages may have a crucial role to play in determining the extent to which tariff liberalisation affects firm-level outcomes.

With regard to India's trade reforms in particular, most studies have tended to focus on tariffs on final goods, or final goods tariffs. However, an increasing body of evidence suggests that declines in tariffs on intermediate inputs (input tariffs) have a greater positive impact on firm-level productivity, relative to final goods tariffs. Amiti and Konings (2007) arrive at this conclusion in a study focusing on Indonesian firms, and Nataraj (2011) obtains a similar result for formal firms in India. This evidence strengthens the case for examining both final goods and input tariff declines in a study of the implications of trade liberalisation for employment.

The only study that appears to have assessed the implications of reductions in input tariffs for informal employment, however, appears to be that of Menezes-Filho and Muendler (2011). This analysis exploits a rich worker flow dataset to establish that final goods and input tariff cuts in Brazil in the 1990s are not associated with significant shifts in informal employment. Empirical analysis otherwise appears to have sidestepped the impacts of input tariff declines on informal employment, a topic which the current study covers.

Several studies, including Goldberg and Pavcnik (2003) and Bosch et al. (2007), suggest that firm-level employment is at least as much a function of the degree of domestic labour market flexibility as it is of economic policy shifts. Intuitively, the notion that the impact of economic reforms on labour markets is affected by domestic institutions is appealing. In other words, the impact of economic reforms on domestic labour markets is arguably likely to hinge on the interaction between policy change and domestic institutions, in particular labour market regulation.

The impacts of labour market regulation on employment outcomes have long constituted an area of research interest. Botero et al. (2004) study labour laws in 85 countries and conclude that more inflexible labour markets (in terms of higher levels of labour regulation) tend to have larger unofficial segments and higher unemployment. Given the federal structure of its economy and the fact that its numerous states (provinces) have considerable autonomy in terms of amending and implementing centrally driven labour market regulation, India offers fertile ground in this context. Besley and Burgess (2004) exploit the state and time-level variation in amendments made to the Industrial Disputes Act (IDA) of 1947 up to 1990 to derive labour market flexibility scores that vary across states and over time (these are discussed in more detail in Section 3). Founded upon these scores, their analysis concludes that states that tended to make more 'pro-worker' amendments over time tended to witness inferior outcomes in terms of employment, output, investment, productivity and urban poverty, relative to states that tended to make more 'pro-employer' amendments over time.

Recent research is supportive of complementarities between the nationwide industry-level reforms undertaken in India and domestic labour market flexibility. Aghion et al. (2008) argue that manufacturing output in states that made more 'pro-worker' amendments as per the Besley–Burgess methodology tended to be lower following the delicensing reforms undertaken in India in the 1990s, relative to states where amendments tended to be 'pro-employer'. Along related lines, Gupta et al. (2009) find that after the delicensing reforms were initiated, states with more inflexible ('pro-worker') labour laws tended to undergo slower employment growth, while states with less competitive product market regulation registered slower output growth. Topalova and Khandelwal (2011), however, use the Besley–Burgess measure to suggest that formal firms in states with more 'pro-worker' legislation experienced higher productivity gains in the wake of India's tariff liberalisation.

A recent study by Hasan et al. (2012) examines the extent to which final goods tariff liberalisation has differential impacts on the unemployment rate in Indian states with relatively more flexible and less flexible labour markets, as evaluated using the Besley–Burgess measure, the measure due to Gupta et al. (2009) and an additional measure ('FLEX 2', described in Section 3). Hasan et al. (2012) conclude that labour market flexibility is conducive to employment growth in the post-liberalisation period, particularly in industries that are net exporters. However, this analysis has limitations. It is conducted at a high level of industry aggregation, does not assess input tariff declines, and does not consider employment in formal and informal enterprises separately. In comparison, the current study focuses on informal enterprises, and uses a more disaggregated industry classification to study both tariff declines and domestic industrial policy reform in India in the 1990s. This paper is therefore an original contribution to the existing evidence base.

Prior to the 1980s, Indian economic policy was largely geared towards government regulation and national self-sufficiency. Trade policy was extremely restrictive and favoured import substitution, with exporters and importers alike facing a wide range of punitive tariff and non-tariff barriers. In tandem, domestic industrial policy imposed several constraints on businesses—most notoriously in the form of

the infamous license policy (the 'License Raj')—and thereby stifled entrepreneurship and growth. Over time, this regulatory regime engendered a productivity decline in the 1970s and became a byword for red tape, graft, inefficiency and government monopoly in several sectors.

In the 1980s, a few reforms were initiated in an attempt to reverse the productivity decline of the previous decade. The domestic license regime was partially liberalised, with roughly one in three three-digit manufacturing industries being delicensed in 1985.[1] In the domain of trade policy, however, tariffs on manufactured imports remained stubbornly high.

The piecemeal reforms of the 1980s proved inadequate in the face of growing fiscal and external macroeconomic imbalances. To worsen matters, a spike in oil prices owing to the Gulf War, a decline in remittance inflows from the Middle East, political uncertainty and a drop in demand for exports to major trade partners all combined to engender substantial capital outflows and, subsequently, a balance-of-payments crisis in 1990–91.

In August 1991, the Indian government approached the IMF to request a Stand-By Arrangement to help it tide over this external payments crisis. The IMF agreed to provide the requisite support conditional on the government undertaking a series of macro-structural reforms, including substantive trade liberalisation measures. It was against this background that the trade reforms of 1991 were phased in. Given the circumstances, it may plausibly be argued that these reforms constituted an exogenous shock for the economy. Sivadasan (2009) and Topalova and Khandelwal (2011) provide additional background detail on the 1991 reforms.

The New Industrial Policy endorsed in 1991 provided a roadmap for reform and the five-year Export Import (Exim) Policy that came into effect in April 1992 encapsulated the new trade policy. Under the trade liberalisation programme initiated in 1992, the import license regime applying to nearly all capital goods and intermediate inputs was abolished. Tariffs were liberalised by capping peak tariff rates and by reducing the number of tariff bands. Furthermore, the Indian rupee was devalued relative to the US dollar, and a dual exchange rate was introduced.

In the 1991–1997 period, the average Indian final goods tariff (ad valorem) on manufactured imports fell from 95 percent to 35 percent (Harrison et al. 2013). However, as Table 1 reveals, the declining trend in final goods tariffs masked considerable dispersion around the mean, with peak tariffs remaining prohibitive. Under the terms of the support extended by the IMF, the deepest tariff cuts were applied to those industries with the highest pre-reform tariff levels. This simplification and harmonisation of the tariff regime was followed by an increase in imports, in particular imports of intermediate inputs.

In 1997, a new five-year Exim Policy was endorsed to consolidate the trade liberalisation and reform process. Tariff reductions continued in the post-1997 period, albeit with less urgency and at a slower pace. Topalova and Khandelwal (2011) argue that endogeneity concerns for this period are likely to be greater relative to the immediate post-reform (1991–1997) period, on the grounds that in contrast to the 1991–1997 period, the later tariff reductions are more likely to have been targeted at protecting less efficient industries. In Section 5.4, I undertake several checks and conclude that tariff endogeneity is unlikely to be a concern for the analysis in this paper.

In tandem, domestic economy deregulation, which had been promoted in 'piecemeal' fashion in the 1980s, received an impetus in the 1990s. This deregulation assumed numerous guises, with the most prominent among them being the quasi-elimination of the notorious industrial license regime and increases in the foreign direct investment (FDI) thresholds applicable to several manufacturing

---

[1] Up until the 1980s, all manufacturing firms with over 50 employees (over 100 employees if electricity was not used) and with assets above a specified threshold were required to obtain a license from the government. This policy was extremely restrictive and discouraged industry entry and competition (Sharma 2008). In this context, the term 'delicensing' implies that firms in a given industry or industries were no longer required to obtain such a license.

industries.[2] On the whole, the reforms of the early nineties resulted in the Indian economy becoming substantially more open relative to its position in the first four decades following independence. As a proportion of GDP, the share of overall trade increased considerably, from 15 percent in the 1980s to about 27 percent in 2000 and further to 47 percent in 2006 (Alessandrini et al. 2011).

As Nataraj (2011) documents, while many of the other domestic reforms of the 1990s were of an industry-invariant nature, the delicensing and FDI liberalisation measures were phased in at different points in time for different manufacturing industries. In all my empirical specifications, I therefore include controls for these reforms (described in Section 3). Other major domestic reforms, such as currency devaluation and corporate tax reform, are accounted for by the use of time fixed effects.

**Table 1.** Summary statistics by year: Final goods tariffs, input tariffs, delicensing and FDI liberalisation (1985–1997) *.

| Year | Final Goods Tariffs (%) | | | | | Input Tariffs (%) | | | | | % DEL | % FDI |
|------|------|--------|--------|-------|-------|------|--------|-------|-------|-------|-------|-------|
|      | Mean | Median | Max | Min | SD | Mean | Median | Max | Min | SD | | |
| **1985** | 88.97 | 91.93 | 203.91 | 0.00 | 32.83 | 57.89 | 58.09 | 86.82 | 23.42 | 11.73 | 35 | 0 |
| **1986** | 95.37 | 100.00 | 242.22 | 0.00 | 37.95 | 60.29 | 59.77 | 88.30 | 23.97 | 11.23 | 36 | 0 |
| **1987** | 94.75 | 100.00 | 242.22 | 0.00 | 37.60 | 58.63 | 58.16 | 79.50 | 23.67 | 10.25 | 36 | 0 |
| **1988** | 94.86 | 100.00 | 248.89 | 0.00 | 37.53 | 59.33 | 58.92 | 83.09 | 23.89 | 10.55 | 36 | 0 |
| **1989** | 95.54 | 100.00 | 281.25 | 0.00 | 40.34 | 59.44 | 59.05 | 83.11 | 23.89 | 10.58 | 37 | 0 |
| **1990** | 95.68 | 100.00 | 281.25 | 0.00 | 40.56 | 59.45 | 58.99 | 83.22 | 23.90 | 10.57 | 37 | 0 |
| **1991** | 95.68 | 100.00 | 281.25 | 0.00 | 40.56 | 59.44 | 58.99 | 83.22 | 23.90 | 10.57 | 84 | 38 |
| **1992** | 63.48 | 64.65 | 281.25 | 0.00 | 27.71 | 39.73 | 40.25 | 53.27 | 20.54 | 5.42 | 84 | 38 |
| **1993** | 63.92 | 64.15 | 340.63 | 22.50 | 31.03 | 38.53 | 39.70 | 54.35 | 20.42 | 5.31 | 86 | 38 |
| **1994** | 64.46 | 65.00 | 400.00 | 11.28 | 36.06 | 37.34 | 37.97 | 55.42 | 8.92 | 6.06 | 86 | 38 |
| **1995** | 53.57 | 53.50 | 320.75 | 12.08 | 30.86 | 30.11 | 30.83 | 48.97 | 8.64 | 5.32 | 86 | 38 |
| **1996** | 42.41 | 44.25 | 254.27 | 0.00 | 24.85 | 22.76 | 23.39 | 42.51 | 8.15 | 5.15 | 86 | 38 |
| **1997** | 34.15 | 34.48 | 176.67 | 0.00 | 18.59 | 18.37 | 19.31 | 32.95 | 6.37 | 4.09 | 89 | 45 |

Source: Final goods and input tariff data obtained from Nataraj (2011); 132 three-digit NIC (1987) industries included
* "% DEL" and "% FDI" refer to the proportions of industries that were delicensed and FDI liberalised (respectively) up to a given year.

## 3. Data

I use survey data on unorganised manufacturing firms compiled by the National Sample Survey Office (NSSO) in 1989–90, 1994–95 and 2000–01. The Factories Act of 1948 requires all Indian manufacturing firms that use electricity and employ 10 or more workers, as well as all manufacturing firms that do not use electricity and employ 20 or more workers, to register with the state government. The term 'workers' encompasses all paid and unpaid individuals, including household help where this is relevant, who are directly or indirectly associated with a firm's operations. All other firms are unregistered and comprise the informal manufacturing sector, which is the sampling frame of the NSSO surveys of unorganised manufacturing enterprises.[3] In the baseline regressions in this paper, employment is captured in terms of the total number of persons engaged. The baseline dependent variable is the natural logarithm of firm-level employment.

---

[2] Prior to 1991, most industries were characterised by a 40 percent FDI ceiling. In 1991 and in the following years, this ceiling was raised to 51 percent for a number of industries, with 'automatic' FDI approval, and other regulations concerning FDI were liberalised (Sivadasan 2009).

[3] The terms 'unorganised' and 'informal' are, in the context of Indian firms, virtually synonymous. In addition to the proprietary and partnership-based enterprises that constitute the informal sector, the unorganised sector also encompasses a small number of enterprises managed by cooperative societies, trusts, and private and public limited companies. As this latter category of enterprises accounts for less than 0.5 percent of the sample of enterprises in each NSSO survey round (sample) used in the analysis for this paper, and as it is arguably unlikely to face the same production or growth incentives as the informal sector, I exclude it from my analysis.

The NSSO surveys approximately 1 percent of all informal enterprises approximately every five years. It employs a stratified random sampling strategy for each survey, with the sample frame in each period updated on the basis of the sample frame used in the preceding Economic Census (EC). I use the inverse sampling probability-based weights that accompany the survey data to weight observations in a manner that yields results that are applicable to the population of small informal enterprises. I observe informal enterprises in one pre-reform period (1990) and two post-reform periods (1995 and 2001). As my data do not comprise a panel, I am unable to pinpoint the channels through which observed employment changes occur, but I discuss this issue to the extent possible.

The pooled employment distribution for the population of informal manufacturers is presented in Figure 1a. The average informal enterprise employs two individuals, a number that displayed remarkable consistency in the 1990s, both in states with flexible and states with inflexible labour markets (defined on the basis of the 'FLEX 2' measure discussed in Section 3), and only registered a very slight decline across the country in 2001, as seen in Figure 1b. Furthermore, over 95 percent of informal enterprises employ fewer than five people. A little over 50 percent of informal manufacturing jobs are accounted for by informal enterprises engaging one or two individuals.

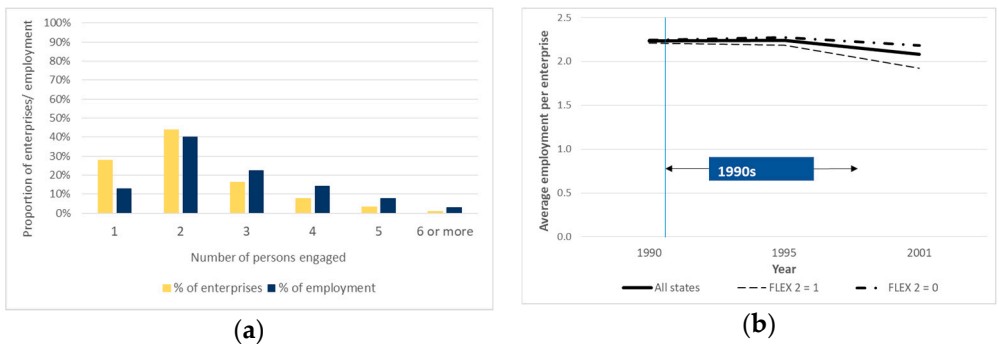

**Figure 1.** Employment in informal manufacturing enterprises in India. (**a**) Informal enterprise and employment shares by enterprise size (1990–2001); (**b**) Average number of persons engaged per informal enterprise (1990–2001). Source: NSSO survey data (1990, 1995, 2001) As inverse sampling probability-based multipliers have been used to aggregate the raw data, these distributions are representative of the population of informal enterprises. The measure of labour market flexibility used in Figure 1b is the 'FLEX 2' measure due to Hasan et al. (2012) and is described in Section 3.

Close to 80 percent of the informal enterprises in my dataset are small, household only enterprises, labelled 'own account manufacturing enterprises' or OAMEs in the NSSO surveys. OAMEs are household-based, informal manufacturing enterprises that do not hire any workers on a regular basis. In effect, OAMEs only employ unpaid members of the household(s) of their proprietor(s). The remaining, slightly larger informal enterprises in the dataset are labelled 'non-directory manufacturing establishments' or NDMEs by the NSSO. NDMEs are informal manufacturing enterprises that hire at least one and up to five workers (household and non-household workers) on a regular basis. In 1990, the NSSO did not survey relatively large informal enterprises employing more than six workers (household and hired workers) on a regular basis (labelled 'directory manufacturing establishments' or DMEs). As DMEs therefore do not feature in the only pre-reform data at my disposal and since they comprise less than ten percent of informal enterprises surveyed in 1995 and 2001, I discard them from the dataset. However, I undertake a robustness check in which they are included in the data (for 1995 and 2001), which yields results that are similar to the baseline (Section 5.4).

The construction of the pooled informal enterprise-level dataset poses several challenges, key among which is the fact that the National Industrial Classification (NIC) system used in the 2001 survey (NIC 1998) differs from that used in the 1990 and 1995 surveys (NIC 1987). In a manner similar to that of Nataraj (2011), I assign each firm in the 2001 dataset to the three-digit NIC 1987 code corresponding to its industry of operation and subsequently map firms to tariff codes on the basis of the concordance

specified by Debroy and Santhanam (1993). This yields a dataset comprising firms operating in 132 three-digit NIC 1987 industries. As the state specific labour market flexibility measure used applies to sixteen states, I discard firms located in most other states. The exception is the national capital region (Delhi), which accounts for a large number of firms relative to the states that are excluded and is assigned an inflexible labour market status in the baseline on account of a lack of relevant data. The baseline results hold if Delhi and Jammu & Kashmir (which is classified as being a state with an inflexible labour market, following the discussion in further in this section) are, instead, assumed to be flexible labour markets (Section 5.4). Restricting the dataset to the sixteen states of interest and Delhi does not appear to be a serious concern, as these regions consistently account for over 95 percent of Indian GDP and, further, the firms retained in my sample account for over 80 percent of informal manufacturing employment in each period.

I exclude informal enterprises that are reported to have been closed from my analysis. Furthermore, I observe that a very small fraction (less than 1 percent) of enterprises in each period appear to employ ten or more persons, the threshold above which units that use electricity attain formal (registered) status. I drop these enterprises from my dataset, but I undertake a robustness check to confirm that the alternative does not affect my baseline results (Section 5.4). Furthermore, I do not include enterprises that report zero or missing values for raw material use and/or physical product manufacturing in the baseline analysis. These enterprises arguably engage solely in trading activity, although they are classified as 'manufacturing' entities. A robustness check suggests that including these 'non-manufacturers' in the analysis does not substantially affect the results (Section 5.4).

I use annual data on final goods and input tariff rates for the 1985–1997 period, compiled by Nataraj (2011) at the three-digit National Industrial Classification (NIC) of 1987 level. The final goods tariff data are based on the Government of India's Customs Tariff Working Schedules and the United Nations Conference on Trade and Development—Trade Analysis Information System (UNCTAD-TRAINS) database, whereas the input tariff data are computed using sectoral final goods tariffs and the Indian Input-Output Transactions Table (IOTT). For example, as explained in Nataraj (2011), if leather goods and textiles comprise 80 percent and 20 percent of the inputs used by the footwear industry, the input tariff for the latter equals 0.8 times the final goods tariff for leather goods plus 0.2 times the final goods tariff for textiles. I follow Harrison et al. (2013) in using input tariffs constructed on the basis of manufacturing and non-manufacturing industry final goods tariffs, and in undertaking a robustness check (discussed in Section 5.4) for which input tariffs constructed using only manufacturing industry final goods tariffs are used. The IOTT classifies industries into only 62 relevant groups as opposed to the NIC (1987) classification, for which over 130 industry codes exist for which final goods tariff data are available. Despite this limitation, a considerable degree of variation is observable in input tariffs across NIC (1987) industries. Summary statistics are provided in Table 1 (Section 2). Final goods and input tariffs are measured in terms of fractions in the dataset (so that, for instance, a tariff rate of 80 percent corresponds to 0.80).

To control for the delicensing and FDI regime reforms undertaken in India in the period of interest, I use industry and time varying indicator variables that are also due to Nataraj (2011). These data were first used by Aghion et al. (2008). The delicensing and FDI reform variables assume a value of '1' for a given industry in a specific year if that industry was delicensed or FDI liberalised by the year in question, and are otherwise equal to '0'. As discussed in Section 2, approximately one-third of three-digit NIC (1987) manufacturing industries (and a little over one-third of the industries represented in my dataset) had been delicensed in 1985. After the 1991 reform episode, the proportion of delicensed industries increased to almost 90 percent, while approximately 40 percent of industries were FDI liberalised.

As discussed in Section 2, final goods tariffs declined precipitously in 1992, which was the first year of reform implementation following the balance-of-payments crisis of 1990–91. Input tariffs also fell and converged in the post-1991 period, and display less variance relative to final goods tariffs. The scatterplots in Figure 2a,b capture tariff levels in 1989 and the declines that occurred in the 1989–2000 period for final goods and input tariffs, illustrating how the highest pre-reform tariff rates

were subjected to the largest cuts.[4] Figure 2c plots pairwise declines in final goods tariffs and input tariffs over the 1989–1994 period. The resulting scatterplot suggests that while there may be a positive association between the shifts in tariff rates, it is not sufficiently strong for multicollinearity to pose major concerns.[5]

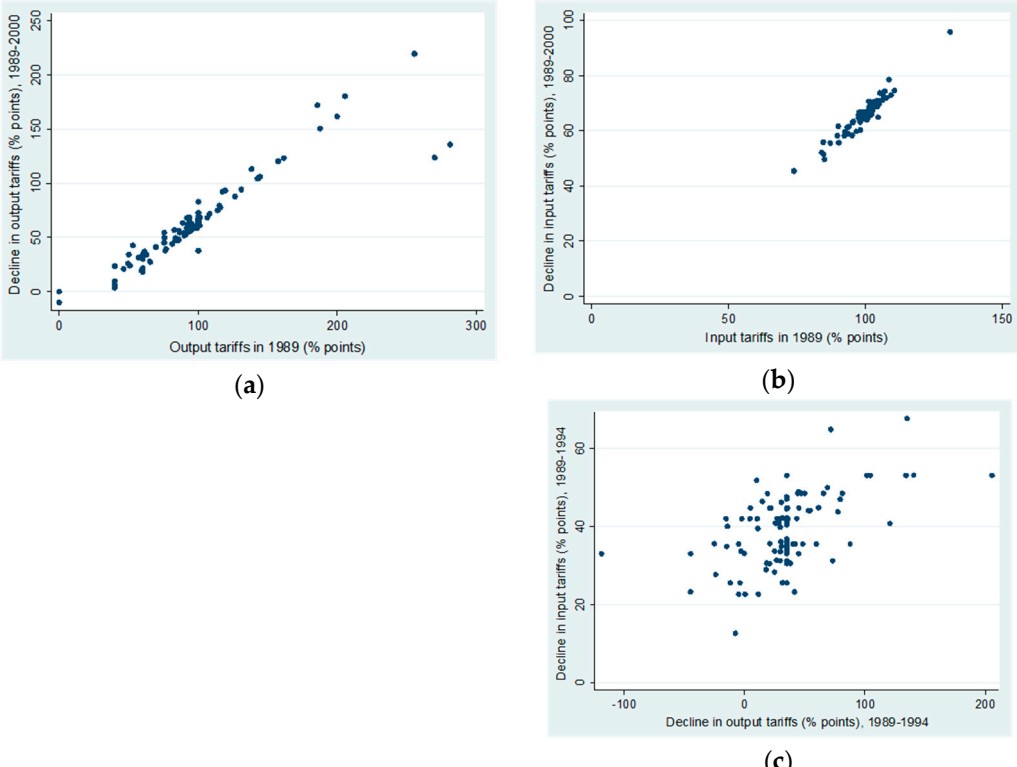

**Figure 2.** Final goods tariffs and input tariffs (1989–2000). (**a**)[6] Final goods tariffs (1989) and declines in final goods tariffs (1989–2000); (**b**) Input tariffs (1989) and declines in input tariffs (1989–2000); (**c**) Declines in final goods tariffs and declines in input tariffs (1989–1994). Source: Final goods and input tariff data compiled by Nataraj (2011) on the basis of Indian government data and India's IOTT.

The measure of state-level labour market flexibility used in this study, labelled 'FLEX 2', is due to Hasan et al. (2012). This measure is founded upon the workhorse measure developed by Besley and Burgess (2004).

Besley and Burgess (2004) use the Industrial Disputes Act (IDA) of 1947, passed by the central government, as their baseline. They exploit the fact that fifteen major Indian states made a series of amendments to this Act in the 1958–1990 period to develop an econometric strategy that accounts for state-level regulatory variation.[7] In total, the fifteen states made 113 amendments. Besley and Burgess assign a code of '1' to each amendment they deem to be 'pro-worker', a code of '−1' to amendments they find to be 'pro-employer' and a code of '0' to 'neutral' amendments. Following this, they assign

---

[4]    This was purposefully undertaken in the case of final goods tariffs, with input tariffs undergoing related, albeit not equivalent, declines.

[5]    The correlation coefficient for the changes in final goods and input tariffs over the 1989–1994 period is 0.5776, while that for the corresponding changes over the 1989–2000 period is 0.5927.

[6]    The two outliers visible to the right of this graph are the wine manufacturing and spirit distillation, rectification and blending industries, the final goods tariffs for which amounted to over 250 percent in 1989, but were subjected to smaller reductions relative to other industries with very high tariff rates in 1989. A robustness check which omits these outliers from the baseline regressions (outlined in Section 5.4) is discussed in Section 5.4.

[7]    Besley and Burgess (2004) consider sixteen states in their analysis, but the state of Jammu and Kashmir made no amendment to the IDA in the 1958–1990 period.

to each state a score of '1', '−1' or '0' in each year when the state passed at least one amendment, based on the dominant direction of amendments passed. For instance, a state which passed three pro-worker amendments ('1+1+1') and one pro-employer amendment ('−1') in 1965 gains a score of one (for having been predominantly pro-worker, in the sense that '1+1+1+(−1)' exceeds zero) for 1965. The year-specific scores assigned to each state are then accumulated over time for all relevant years (years in which the state made at least one amendment) to arrive at a final state-specific score for 1990, on the basis of which the state is classified as being pro-worker, pro-employer or neutral in any given year.

Gupta et al. (2009) modify the Besley–Burgess measure to account for several suggestions offered by Bhattacharjea (2006) and for Organization for Economic Co-Operation and Development (2007) survey research that assesses areas in which states have undertaken measures pertinent to the implementation of labour laws (including but not limited to the IDA). The labour market flexibility indicator developed by Gupta et al. (2009) is labelled 'FLEX 3' by Hasan et al. (2012), who construct an additional measure that they refer to as 'FLEX 2'.[8] Also rooted in the Besley–Burgess measure, the 'FLEX 2' index inverts the final Besley–Burgess scores of three states: Gujarat, Kerala and Maharashtra. Hasan et al. point out that World Bank (2005) research supports the view that Gujarat and Maharashtra, assigned overall scores of '1' (pro-worker status) by Besley and Burgess, are generally regarded favourably by business representatives, whereas Kerala, although designated to be pro-employer by Besley and Burgess, is perceived to have a 'poor investment climate'.[9] In summary, the 'FLEX 2' index assigns scores of −1, −1 and 1 to Gujarat, Maharashtra and Kerala respectively. Table 2 summarises the 'FLEX 1' (Besley and Burgess' index), 'FLEX 2' and 'FLEX 3' scores for each state.

In this study, I use the 'FLEX 2' measure of labour market flexibility, as it takes account not only of the nature of labour market regulation but also of business managers' perceptions regarding the enforcement of the same in terms of state specific investment environments (see footnote 10). Dougherty (2009) notes that there were no major state-level amendments to the IDA between 1990 and 2004.[10] As my analysis is focused on the 1990–2001 time period, the 'FLEX 2' indicator varies only across states and not over time.

As I interact the 'FLEX 2' measure with the final goods and input tariffs in my regressions, I recode the 'FLEX 2' index to facilitate the interpretation of my findings. Along the lines of Hasan et al. (2012), states with flexible ('pro-employer') labour markets receive a score of '1' (rather than '−1', as is the case in the Besley–Burgess scores), whereas states with neutral or inflexible ('pro-worker') labour markets receive a score of '0' (rather than '1' for the states with inflexible labour laws, as is the case in the Besley–Burgess index).

Table 3 provides summary employment statistics for OAMEs and NDMEs, for the sample as a whole and separately for states with flexible and inflexible labour markets, as defined using the 'FLEX 2' measure. The average OAME engages two people, while three to four individuals are engaged in

---

[8]    The Besley–Burgess measure, with a minor correction incorporated for the state of Madhya Pradesh, is labelled 'FLEX 1' by Hasan et al. (2012).

[9]    In their online appendix, Hasan et al. (2012) provide additional detail in this regard. Gujarat and Maharashtra are typically considered to be prime business locations by Indian businessmen, whereas Kerala is not. The World Bank's (2005) research presents firm level survey findings in which managers rank Maharashtra and Gujarat highly, labelling them to be 'Best Investment Climate' states more consistently than other states. Kerala, conversely, attains a 'Poor Investment Climate' ranking. Small and medium-sized firms report having been subjected to twice as many factory inspections in 'Poor Investment Climate' states as in 'Best Investment Climate' states, suggesting that enforcement of ostensibly 'pro-worker' amendments to the IDA is likely to be less stringent in the latter type of state. Further, firms perceive that 'over-manning' (the gap between optimal and actual employment levels given current output levels) is on average less visible in Maharashtra and Gujarat than elsewhere. In 'Poor Investment Climate' states (such as Kerala), restrictive labour regulations were considered to be a primary driver of 'over-manning', whereas in 'Best Investment Climate' states, 'over-manning' (lower than in other states in the first place) was perceived more favourably, in the sense that it was considered to occur when firms expected higher future growth.

[10]    Dougherty (2009) states that there have been only eight state-level IDA amendments in the post-1990 period, of which the only amendments of relevance for labour market outcomes were made by the state of Gujarat in 2004, which falls outside the period of interest for my analysis.

the average NDME. Only minor differences appear in these numbers across the two groups of states. While both averages register declines in 2001 relative to 1990, these changes are very small and do not, prima facie, appear to be economically meaningful.

**Table 2.** Summary of labour market flexibility indices *.

| State | Measure of Labour Market Flexibility * | | |
|---|---|---|---|
| | FLEX 1 | FLEX 2 | FLEX 3 |
| Andhra Pradesh | 1 | 1 | 1 |
| Assam | 0 | 0 | 0 |
| Bihar | 0 | 0 | 0 |
| Gujarat | 0 | 1 | 0 |
| Haryana | 0 | 0 | 0 |
| Karnataka | 1 | 1 | 1 |
| Kerala | 1 | 0 | 0 |
| Madhya Pradesh | 0 | 0 | 0 |
| Maharashtra | 0 | 1 | 0 |
| Orissa | 0 | 0 | 0 |
| Punjab | 0 | 0 | 0 |
| Rajasthan | 1 | 1 | 1 |
| Tamil Nadu | 1 | 1 | 1 |
| Uttar Pradesh | 0 | 0 | 1 |
| West Bengal | 0 | 0 | 0 |

Source: Besley and Burgess (2004)—FLEX 1; Hasan et al. (2012)—FLEX 2; Gupta et al. (2009)—FLEX 3. * Recoded scores: 1 = flexible labour market regulation, 0 = inflexible labour market regulation.

**Table 3.** Summary statistics for employment in informal enterprises (1990–2001).

| | OAMEs | | | | NDMEs | | | |
|---|---|---|---|---|---|---|---|---|
| Year | N | Mean | St. dev. | Weighted Total * | N | Mean | St. dev. | Weighted Total * |
| | | | | Overall | | | | |
| 1990 | 29,661 | 2.12 | 1.10 | 12,409,060 | 11,428 | 3.55 | 1.15 | 1,836,999 |
| 1995 | 87,410 | 2.18 | 1.12 | 15,836,887 | 19,146 | 3.40 | 1.19 | 3,554,920 |
| 2001 | 45,305 | 1.81 | 0.94 | 12,715,694 | 16,858 | 3.35 | 1.11 | 3,029,317 |
| **Overall** | **162,376** | **2.06** | **1.08** | **40,961,641** | **47,432** | **3.42** | **1.15** | **8,421,235** |
| | | | States with flexible labour markets (FLEX 2 = 1) | | | | | |
| 1990 | 10,868 | 2.11 | 1.08 | 3,475,971 | 4406 | 3.68 | 1.13 | 748,955 |
| 1995 | 35,327 | 2.14 | 1.10 | 5,136,175 | 8744 | 3.55 | 1.20 | 1,561,064 |
| 2001 | 18,576 | 1.70 | 0.91 | 4,299,569 | 6560 | 3.43 | 1.12 | 1,311,631 |
| **Overall** | **64,771** | **2.01** | **1.06** | **12,911,715** | **19,710** | **3.54** | **1.16** | **3,621,650** |
| | | | States with inflexible labour markets (FLEX 2 = 0) | | | | | |
| 1990 | 18,793 | 2.12 | 1.11 | 8,933,089 | 7022 | 3.46 | 1.15 | 1,088,044 |
| 1995 | 52,083 | 2.20 | 1.13 | 10,700,713 | 10,402 | 3.28 | 1.16 | 1,993,856 |
| 2001 | 26,729 | 1.89 | 0.96 | 8,416,125 | 10,298 | 3.30 | 1.10 | 1,717,685 |
| **Overall** | **97,605** | **2.10** | **1.09** | **28,049,926** | **27,722** | **3.33** | **1.14** | **4,799,585** |

Source: NSSO data (1990, 1995, 2001) The data are unweighted and apply only to the sample of informal enterprises surveyed in each year. N: number of observations; St. dev.: standard deviation (the minimum and maximum numbers for each row in this table are 1 and 9 respectively). * This refers to the total number of persons engaged in the population represented by the informal enterprises in the sample dataset, derived using the survey weights provided for each enterprise surveyed in the sample dataset.

The final column of Table 3 also shows that, overall, total weighted employment in the population represented by OAMEs and NDMEs increased over the 1990–2001 period, both in states with more flexible labour markets and less flexible labour markets. More precisely, the data are indicative of a

substantial increase in aggregate employment in informal enterprises in all states in the 1990–1995 period, followed by a slight decline in the 1995–2001 period. However, the 2001 figures are clearly higher than the corresponding 1990 estimates in each case, with the exception of OAMEs in states with less flexible labour markets. Considered alongside the observed declines on average informal enterprise employment in this period, these data highlight that a more aggregated, industry-level analysis of the impacts of the reforms on informal enterprise numbers and employment may be of relevance for this paper. Following the strategy employed in Section 4, I also attempt to disentangle the implications of the varying policy shifts of the 1990s for employment in the informal sector at the 'micro' (enterprise) and 'macro' (industry) levels.

## 4. Methodology

The analysis harnesses the variation in policy change over time and across industries in India in the 1990s, as outlined in Sections 2 and 3, to identify the impact of economic reform on employment. In the expanded baseline specification, I account for state-level differences in labour market flexibility.

The preliminary regression that I employ is of the form:

$$\mathrm{lnemp}_{ijkt} = \alpha_0 + \alpha_1 \mathrm{TAR}_{jt-2} + \alpha_2 \mathrm{INT}_{jt-2} + \alpha_3 \mathrm{DEL}_{jt-2} + \alpha_4 \mathrm{FDI}_{jt-2} + \delta_t + \delta_j + \delta_k + \varepsilon_{ijkt}, \tag{1}$$

where $\ln(\mathrm{emp})_{ijkt}$ is the natural logarithm of paid employment in enterprise i in industry j and state k at time t; $\mathrm{TAR}_{jt-2}$ and $\mathrm{INT}_{jt-2}$ are two-year lags of final goods and input tariffs; $\mathrm{DEL}_{jt-2}$ and $\mathrm{FDI}_{jt-2}$ are time-varying indicator variables capturing whether industry j underwent delicensing and FDI regime reforms two years prior to year t; and $\delta_t$, $\delta_j$ and $\delta_k$ are year, industry and state fixed effects. To explore any overarching associations between the reforms and average enterprise-level employment, irrespective of variations in state-level flexibility, I use Equation (1) as a primary enterprise-level specification. As most informal enterprises employ less than five individuals, following the discussion in Section 3, I also explore whether the results yielded by the baseline specification hold when a Poisson count model is adopted. The dependent variable for the Poisson regressions is the actual number of individuals engaged in work in a given enterprise, as opposed to the natural logarithm of that number. Standard errors in all the enterprise-level regressions are clustered at the state-industry level to avoid heteroscedasticity issues (with Huber-White standard errors having been found to yield virtually the same findings).

I also use a variant of Equation (1) to undertake panel fixed effects analysis at a broader, three-digit industry level, for the economy as a whole as well as separately for states with flexible and inflexible labour markets. This analysis, discussed in Section 5.2, considers the implications of the reforms for the 'extensive margins' of enterprise numbers and aggregate employment at the industry level (in logarithms). These industry-level regressions are weighted by the pre-reform (1990) industry levels of the dependent variable in each case. Following Martin et al. (2017), this analysis is restricted to industries that have ten or more enterprises in each weighted cross-section, a step which omits only a small number of industries.

The expanded baseline specification that I use to examine the implications of differences in state-level labour market flexibility is similar to that used by Hasan et al. (2012):

$$\begin{aligned}
\mathrm{lnemp}_{ijkt} = {} & \alpha_0 + \alpha_1 \mathrm{TARIFF}_{jt-2} + \beta_1 \mathrm{TARIFF}_{jt-2} \mathrm{LM}_k + \alpha_2 \mathrm{INTAR}_{jt-2} + \\
& \beta_2 \mathrm{INTAR}_{jt-2} \mathrm{LM}_k + \alpha_3 \mathrm{DEL}_{jt-2} + \beta_3 \mathrm{DEL}_{jt-2} \mathrm{LM}_k + \alpha_4 \mathrm{FDI}_{jt-2} + \beta_4 \mathrm{FDI}_{jt-2} \mathrm{LM}_k + \delta_t + \\
& \delta_j + \delta_k + \varepsilon_{ijkt},
\end{aligned} \tag{2}$$

where $\mathrm{LM}_k$ is a time-invariant indicator variable capturing the degree of labour market flexibility in state k (the 'FLEX 2' measure), and the other variables follow the description provided for Equation (1). As $\mathrm{LM}_k$ is time invariant, its level effect is subsumed within $\delta_k$, the state fixed effects term.

Effectively, both the firm and industry-level specifications, outlined in Equations (1) and (2), are difference-in-differences regressions. This strategy appears valid in this context, given that the reforms in question affected different industries to varying degrees at different points in time. At the enterprise level, the key limitation of my estimation strategy lies in the lack of panel data, which restricts my results to being net effects. In addition, I cannot track informal enterprises exiting and entering an industry. However, at the industry level, I am able to use panel data, as described above. A wider limitation of the dataset arises from the fact that the NSSO surveys informal enterprises only once in every five years, which means that the repeated cross-sections represented in my dataset are somewhat spaced out (1990, 1995 and 2001). Nonetheless, this is the best dataset available in the context of the research question.

In the specification presented in Equation (1), the overall impact of the reforms on employment is the sum of the coefficients $\alpha_1$, $\alpha_2$, $\alpha_3$ and $\alpha_4$. In the expanded specification of Equation (2), this impact derives from the sums $\alpha_1 + \beta_1 LM_k$ (for final goods tariff liberalisation), $\alpha_2 + \beta_2 LM_k$ (for input tariff liberalisation), $\alpha_3 + \beta_3 LM_k$ (for delicensing) and $\alpha_4 + \beta_4 LM_k$ (for FDI reform). In each instance, the first term captures the direct impact linked with the reform in question, whereas the interaction term (involving $LM_k$) presents a measure of the indirect effect associated with the interplay between the reform and state-level labour market flexibility. The sum of the two coefficients thus yields a measure of the net impact of each reform measure on average enterprise-level employment. This varies across states, with the interaction-based effect amounting to zero for states with inflexible labour markets (as the 'FLEX 2' variable equals zero for these states).

As discussed in Hasan et al. (2012), significant interstate migration flows could pose a threat to my identification strategy, by resulting in overestimation of the β coefficients. Although my tariff measures are state invariant, it could be argued that substantial tariff declines might result in larger numbers of workers moving out of states with more flexible labour markets, relative to states with less flexible labour markets. However, as Hasan et al. (2012) document, work undertaken by Dyson et al. (2004); Anant et al. (2006); Munshi and Rosenzweig (2009); and Topalova (2010) suggests that migration within India has tended to be insubstantial in recent decades, with interstate migration levels having been particularly low. This indicates that any worker flows engendered by the trade reforms were limited, with spillovers straddling state borders likely to have been rare.

## 5. Results

### 5.1. Baseline Regressions: Enterprise Level

To begin, I assess whether the reforms are associated with statistically significant employment shifts at the enterprise level, irrespective of variations in regional labour market flexibility. In Table 4, I therefore run variations of Equation (1) presented in Section 4. Neither final goods nor input tariff reductions are associated with significant employment changes in informal enterprises. However, I find that the delicensing reform is associated with a statistically significant increase on average informal enterprise-level employment. This result is robust to controlling for FDI reform, which shows no significant link with enterprise-level employment. Specifically, controlling for final goods and input tariff declines, FDI liberalisation, and state, year and industry fixed effects, I find that delicensing is associated with employment in the average informal enterprise rising by 7.6 percent (Table 4, Column 5).[11] In this last specification, input tariff reductions are associated with a decrease in informal enterprise-level employment, but this result is only weakly statistically significant.

---

[11]　As specified in Section 3, all the tariffs are entered into the dataset in fractional form (for instance, a tariff of 80 percent is entered as 0.80). As a result, given that the dependent variable is in logarithmic form, we may interpret any coefficients attaching to the tariffs as proportional changes directly (without having to multiply them by 100). On the other hand, as the delicensing and FDI reform variables are indicator variables and cannot be rescaled in a manner similar to the tariffs, the coefficients that attach to them must be multiplied by 100 for appropriate interpretation (given the logarithmic form of the dependent variable).

**Table 4.** Economic reforms and employment in informal enterprises (1990–2001): OLS estimates.

|  | (1) | (2) | (3) | (4) | (5) |
|---|---|---|---|---|---|
| Final goods tariffs | −0.011 | −0.082 | −0.090 | −0.082 | −0.091 |
|  | (0.086) | (0.108) | (0.102) | (0.109) | (0.104) |
| Input tariffs |  | 0.473 | 0.550 * | 0.472 | 0.558 * |
|  |  | (0.314) | (0.314) | (0.316) | (0.317) |
| Delicensing |  |  | 0.068 ** |  | 0.076 *** |
|  |  |  | (0.026) |  | (0.028) |
| FDI reform |  |  |  | 0.019 | 0.041 |
|  |  |  |  | (0.026) | (0.026) |
| State FE | Yes | Yes | Yes | Yes | Yes |
| Year FE | Yes | Yes | Yes | Yes | Yes |
| Industry FE | Yes | Yes | Yes | Yes | Yes |
| Observations | 195,789 | 195,789 | 195,789 | 195,789 | 195,789 |
| R-squared | 0.160 | 0.161 | 0.162 | 0.161 | 0.162 |

Dependent variable: natural logarithm of total number of persons engaged; 'FE': fixed effects. Standard errors, in brackets, are clustered at the state-industry level. ***: Significant at 1%, **: Significant at 5%, *: Significant at 10%.

In Table 5, I explore the extent to which state-level differences in labour market flexibility have a bearing on the effects of the reforms, using alternative forms of the expanded baseline specification of Equation (2) discussed in Section 4. I focus on the results that are statistically significant at the significance level of 0.05. First, I confirm that final goods and input tariff reductions are not associated with significant changes in employment in all states, with the weakly significant negative effect for input tariff declines, visible in Table 4, being restricted to states with inflexible labour markets (Table 5, Row 3).

**Table 5.** Economic reforms, labour market flexibility and employment in informal enterprises (1990–2001): OLS estimates.

|  | (1) | (2) | (3) | (4) | (5) | (6) |
|---|---|---|---|---|---|---|
| Final goods tariffs | −0.030 | −0.055 | −0.030 | −0.138 | −0.137 | −0.132 |
|  | (0.092) | (0.092) | (0.091) | (0.115) | (0.118) | (0.113) |
| Final goods tariffs * FLEX 2 | 0.056 | 0.140 ** | 0.076 | 0.143 | 0.147 | 0.151 |
|  | (0.063) | (0.058) | (0.063) | (0.098) | (0.103) | (0.097) |
| Input tariffs |  |  |  | 0.599 * | 0.538 * | 0.576 * |
|  |  |  |  | (0.328) | (0.323) | (0.327) |
| Input tariffs * FLEX 2 |  |  |  | −0.202 | −0.022 | −0.178 |
|  |  |  |  | (0.172) | (0.166) | (0.168) |
| Delicensing | 0.098 *** | 0.072 *** | 0.100 *** | 0.106 *** | 0.077 *** | 0.108 *** |
|  | (0.029) | (0.027) | (0.029) | (0.030) | (0.027) | (0.030) |
| Delicensing * FLEX 2 | −0.070 ** |  | −0.078 ** | −0.083 ** |  | −0.088 ** |
|  | (0.035) |  | (0.035) | (0.037) |  | (0.037) |
| FDI reform | 0.039 | 0.022 | 0.019 | 0.042 | 0.024 | 0.024 |
|  | (0.026) | (0.028) | (0.028) | (0.026) | (0.028) | (0.028) |
| FDI reform * FLEX 2 |  | 0.069 * | 0.082 ** |  | 0.065 | 0.076 * |
|  |  | (0.041) | (0.041) |  | (0.041) | (0.041) |
| **Flexible labour markets: Effects of changes in final goods tariffs** |  |  |  |  |  |  |
| (Final goods tariffs)*(1 + FLEX 2) | 0.026 | 0.085 | 0.047 | 0.004 | 0.010 | 0.019 |
| Standard Error | 0.069 | 0.067 | 0.068 | 0.079 | 0.083 | 0.078 |
| p-value (combined effect = 0) | 0.711 | 0.208 | 0.494 | 0.957 | 0.907 | 0.809 |

**Table 5.** *Cont.*

| | (1) | (2) | (3) | (4) | (5) | (6) |
|---|---|---|---|---|---|---|
| **Flexible labour markets: Effects of changes in input tariffs** | | | | | | |
| **(Input tariffs)*(1 + FLEX 2)** | | | | 0.397 | 0.516 * | 0.398 |
| Standard Error | | | | 0.300 | 0.305 | 0.300 |
| *p*-value (combined effect = 0) | | | | 0.186 | 0.091 | 0.185 |
| **Flexible labour markets: Effects of delicensing** | | | | | | |
| **(Delicensing)*(1 + FLEX 2)** | 0.028 | | 0.022 | 0.024 | | 0.019 |
| Standard Error | 0.038 | | 0.038 | 0.038 | | 0.038 |
| *p*-value (combined effect = 0) | 0.468 | | 0.563 | 0.536 | | 0.613 |
| **Flexible labour markets: Effects of FDI reform** | | | | | | |
| **(FDI reform)*(1 + FLEX 2)** | | 0.090 ** | 0.101 ** | | 0.090 ** | 0.099 ** |
| Standard Error | | 0.043 | 0.043 | | 0.043 | 0.043 |
| *p*-value (combined effect = 0) | | 0.036 | 0.018 | | 0.036 | 0.021 |
| State FE | Yes | Yes | Yes | Yes | Yes | Yes |
| Year FE | Yes | Yes | Yes | Yes | Yes | Yes |
| Industry FE | Yes | Yes | Yes | Yes | Yes | Yes |
| Observations | 195,789 | 195,789 | 195,789 | 195,789 | 195,789 | 195,789 |
| R-squared | 0.163 | 0.163 | 0.164 | 0.164 | 0.163 | 0.164 |

Dependent variable: natural logarithm of total number of persons engaged; 'FE': fixed effects. Standard errors, in brackets, are clustered at the state-industry level. ***: Significant at 1%, **: Significant at 5%, *: Significant at 10%.

The employment enhancing effect of delicensing, visible in Table 4, is restricted to states with inflexible labour markets in Table 5. More precisely, controlling for the other reforms, in states with inflexible labour markets, delicensing is associated with average informal enterprise employment rising by 10.8 percent (Table 5, Column 6, Row 5). This result is statistically significant even at the 0.01 significance level. In states with flexible labour markets, however, the delicensing effect, although still positive, loses statistical significance given the *p*-value of 0.613 (Table 5, Column 6, '(Delicensing)*(1 + FLEX 2)').

Interestingly, Table 5 also reveals that labour market flexibility appears to matter in terms of the response of average informal enterprise-level employment to FDI reform. In states with inflexible labour markets, FDI liberalisation is not associated with statistically significant changes in informal enterprise-level employment, on average and ceteris paribus (Table 5, Column 5, Row 7). Conversely, in states with flexible labour markets, FDI liberalisation is associated with employment in informal enterprises being significantly higher ceteris paribus, by an average of 9.9 percent (Table 5, Column 6, '(FDI reform)*(1 + FLEX 2)').

As discussed in Section 4, I use a Poisson count specification to explore whether the results presented in Tables 4 and 5 are excessively reliant on the fact that the baseline regressions adopt the natural logarithm of employment to capture employment shifts in the tiny informal enterprises in my dataset. The results of this strategy, presented in Tables 6 and 7, provide reassuring evidence that this is not the case. Indeed, the Poisson model yields findings that are very similar, in terms of direction and statistical significance, to the baseline results.

**Table 6.** Economic reforms and employment in informal enterprises (1990–2001): Poisson estimates.

|  | (1) | (2) | (3) | (4) | (5) |
|---|---|---|---|---|---|
| Final goods tariffs | 0.000 | −0.000 | −0.001 | −0.000 | −0.001 |
|  | (0.001) | (0.001) | (0.001) | (0.001) | (0.001) |
| Input tariffs |  | 0.004 | 0.004 | 0.004 | 0.004 |
|  |  | (0.003) | (0.003) | (0.003) | (0.003) |
| Delicensing |  |  | 0.060 ** |  | 0.067 ** |
|  |  |  | (0.026) |  | (0.027) |
| FDI reform |  |  |  | 0.016 | 0.034 |
|  |  |  |  | (0.027) | (0.026) |
| State FE | Yes | Yes | Yes | Yes | Yes |
| Year FE | Yes | Yes | Yes | Yes | Yes |
| Industry FE | Yes | Yes | Yes | Yes | Yes |
| Observations | 195,789 | 195,789 | 195,789 | 195,789 | 195,789 |

Dependent variable: total number of persons engaged; 'FE' denotes fixed effects. Standard errors, in brackets, are clustered at the state-industry level. **: Significant at 5%.

To summarise, the baseline results suggest that on the whole, employment in India's informal manufacturing enterprises in the 1990s responded primarily to the delicensing and FDI reforms of that period, and did not register significant changes in response to the concurrent declines in final goods and input tariffs. Delicensing is associated with significant increases (no significant change) on average informal enterprise-level employment in states with inflexible (flexible) labour markets, while FDI reform is associated with significant increases (no significant change) on average informal enterprise-level employment in states with flexible (inflexible) labour markets.

The distinction between small, household only enterprises (OAMEs) and slightly larger informal enterprises (NDMEs), outlined in Section 3, suggests that the baseline regressions should be separately undertaken for these two firm types. This is also relevant in light of the consideration that enterprise heterogeneity within the informal sector is likely to matter if the impact of the reforms on informal enterprises 'spills over' through the formal sector, as outlined in Section 1. Table 8 presents the findings of this exercise, using both the baseline OLS regression (2) and its Poisson counterpart. The employment enhancing effect associated with delicensing in states with inflexible labour markets is robust for OAMEs, the tiny informal enterprises that dominate the informal sector, but is only weakly significant for the slightly larger NDMEs. Conversely, the baseline increase on average informal enterprise-level employment linked to FDI reform holds primarily for NDMEs, with statistical significance weakening to over 11 percent for OAMEs.

**Table 7.** Economic reforms, labour market flexibility and employment in informal enterprises (1990–2001): Poisson estimates.

| | (1) | (2) | (3) | (4) | (5) | (6) |
|---|---|---|---|---|---|---|
| Final goods tariffs | −0.000 | −0.000 | −0.000 | −0.001 | −0.001 | −0.001 |
| | (0.001) | (0.001) | (0.001) | (0.001) | (0.001) | (0.001) |
| Final goods tariffs * FLEX 2 | 0.000 | 0.001 ** | 0.001 | 0.001 | 0.001 | 0.001 |
| | (0.001) | (0.001) | (0.001) | (0.001) | (0.001) | (0.001) |
| Input tariffs | | | | 0.005 | 0.004 | 0.004 |
| | | | | (0.003) | (0.003) | (0.003) |
| Input tariffs * FLEX 2 | | | | −0.002 | 0.000 | −0.001 |
| | | | | (0.002) | (0.002) | (0.002) |
| Delicensing | 0.087 *** | 0.064** | 0.089 *** | 0.093 *** | 0.067 ** | 0.094 *** |
| | (0.030) | (0.027) | (0.030) | (0.031) | (0.027) | (0.030) |
| Delicensing * FLEX 2 | −0.063 * | | −0.072 ** | −0.073 ** | | −0.080 ** |
| | (0.035) | | (0.035) | (0.037) | | (0.037) |
| FDI reform | 0.033 | 0.014 | 0.012 | 0.035 | 0.015 | 0.014 |
| | (0.026) | (0.027) | (0.027) | (0.026) | (0.027) | (0.028) |
| FDI reform * FLEX 2 | | 0.077 ** | 0.090 ** | | 0.077 ** | 0.086 ** |
| | | (0.039) | (0.039) | | (0.038) | (0.038) |
| **Flexible labour markets: Effects of changes in final goods tariffs** | | | | | | |
| **(Final goods tariffs)*(1 + FLEX 2)** | 0.000 | 0.001 | 0.001 | 0.000 | 0.000 | 0.000 |
| Standard Error | 0.001 | 0.001 | 0.001 | 0.001 | 0.001 | 0.001 |
| *p*-value (combined effect = 0) | 0.654 | 0.171 | 0.398 | 0.889 | 0.827 | 0.740 |
| **Flexible labour markets: Effects of changes in input tariffs** | | | | | | |
| **(Input tariffs)*(1 + FLEX 2)** | | | | 0.003 | 0.004 | 0.003 |
| Standard Error | | | | 0.003 | 0.003 | 0.003 |
| *p*-value (combined effect = 0) | | | | 0.269 | 0.144 | 0.251 |
| **Flexible labour markets: Effects of delicensing** | | | | | | |
| **(Delicensing)*(1 + FLEX 2)** | 0.024 | | 0.017 | 0.019 | | 0.014 |
| Standard Error | 0.036 | | 0.036 | 0.035 | | 0.035 |
| *p*-value (combined effect = 0) | 0.504 | | 0.635 | 0.580 | | 0.682 |
| **Flexible labour markets: Effects of FDI reform** | | | | | | |
| **(FDI reform)*(1 + FLEX 2)** | | 0.091 ** | 0.102 ** | | 0.092 ** | 0.100 ** |
| Standard Error | | 0.041 | 0.041 | | 0.041 | 0.040 |
| *p*-value (combined effect = 0) | | 0.026 | 0.012 | | 0.024 | 0.013 |
| State FE | Yes | Yes | Yes | Yes | Yes | Yes |
| Year FE | Yes | Yes | Yes | Yes | Yes | Yes |
| Industry FE | Yes | Yes | Yes | Yes | Yes | Yes |
| Observations | 195,789 | 195,789 | 195,789 | 195,789 | 195,789 | 195,789 |

Dependent variable: total number of persons engaged; 'FE' denotes fixed effects. Standard errors, in brackets, are clustered at the state-industry level. ***: Significant at 1%, **: Significant at 5%, *: Significant at 10%.

**Table 8.** Economic reforms, labour market flexibility and employment in informal enterprises (1990–2001): Results by firm size (OAME/NDME).

| | OLS Estimates | | | Poisson Estimates | | |
|---|---|---|---|---|---|---|
| | Baseline (All Enterprises) | OAME | NDME | Baseline (All Enterprises) | OAME | NDME |
| Final goods tariffs | −0.132 | −0.179 | 0.005 | −0.001 | −0.001 | 0.000 |
| | (0.113) | (0.133) | (0.046) | (0.001) | (0.001) | (0.000) |
| Final goods tariffs * FLEX 2 | 0.151 | 0.223 * | −0.029 | 0.001 | 0.002 * | −0.000 |
| | (0.097) | (0.115) | (0.059) | (0.001) | (0.001) | (0.001) |
| Input tariffs | 0.576 * | 0.712 * | −0.162 | 0.004 | 0.007* | −0.003 |
| | (0.327) | (0.366) | (0.206) | (0.003) | (0.004) | (0.002) |
| Input tariffs * FLEX 2 | −0.178 | −0.376 ** | 0.089 | −0.001 | −0.004 * | 0.001 |
| | (0.168) | (0.191) | (0.110) | (0.002) | (0.002) | (0.001) |
| Delicensing | 0.108 *** | 0.115 *** | 0.056 * | 0.094 *** | 0.106 *** | 0.040 |
| | (0.030) | (0.032) | (0.029) | (0.030) | (0.034) | (0.027) |
| Delicensing * FLEX 2 | −0.088 ** | −0.116 *** | −0.049 | −0.080** | −0.115 *** | −0.031 |
| | (0.037) | (0.037) | (0.032) | (0.037) | (0.038) | (0.028) |
| FDI reform | 0.024 | 0.048 | 0.036 | 0.014 | 0.038 | 0.035 |
| | (0.028) | (0.031) | (0.029) | (0.028) | (0.032) | (0.026) |
| FDI reform * FLEX 2 | 0.076 * | 0.026 | 0.055 ** | 0.086 ** | 0.034 | 0.048 ** |
| | (0.041) | (0.046) | (0.024) | (0.038) | (0.048) | (0.023) |
| **Flexible labour markets: Effects of changes in final goods tariffs** | | | | | | |
| (Final goods tariffs)*(1 + FLEX 2) | 0.019 | 0.044 | −0.025 | 0.000 | 0.001 | −0.000 |
| Standard Error | 0.078 | 0.101 | 0.051 | 0.001 | 0.001 | 0.000 |
| *p*-value (combined effect = 0) | 0.809 | 0.667 | 0.626 | 0.740 | 0.539 | 0.583 |
| (Input tariffs)*(1 + FLEX 2) | 0.398 | 0.336 | −0.073 | 0.003 | 0.003 | −0.002 |
| Standard Error | 0.300 | 0.330 | 0.220 | 0.003 | 0.003 | 0.002 |
| *p*-value (combined effect = 0) | 0.185 | 0.308 | 0.740 | 0.251 | 0.355 | 0.426 |
| (Delicensing)*(1 + FLEX 2) | 0.019 | −0.001 | 0.008 | 0.014 | −0.009 | 0.009 |
| Standard Error | 0.038 | 0.038 | 0.029 | 0.035 | 0.036 | 0.026 |
| *p*-value (combined effect = 0) | 0.613 | 0.989 | 0.795 | 0.682 | 0.804 | 0.736 |
| **Flexible labour markets: Effects of FDI reform** | | | | | | |
| (FDI reform)*(1 + FLEX 2) | 0.099 ** | 0.074 | 0.091 *** | 0.100 ** | 0.072 | 0.083 *** |
| Standard Error | 0.043 | 0.047 | 0.031 | 0.040 | 0.049 | 0.029 |
| *p*-value (combined effect = 0) | 0.021 | 0.114 | 0.004 | 0.013 | 0.143 | 0.004 |
| State FE | Yes | Yes | Yes | Yes | Yes | Yes |
| Year FE | Yes | Yes | Yes | Yes | Yes | Yes |
| Industry FE | Yes | Yes | Yes | Yes | Yes | Yes |
| Observations | 195,789 | 152,178 | 43,611 | 195,789 | 152,178 | 43,611 |
| R-squared | 0.164 | 0.191 | 0.128 | | | |

Dependent variable: natural logarithm of total number of persons engaged (Columns 1, 2, 3) and total number of persons engaged (Columns 4, 5, 6); 'FE' denotes fixed effects. Standard errors, in brackets, are clustered at the state-industry level. ***: Significant at 1%, **: Significant at 5%, *: Significant at 10%.

## 5.2. Industry-Level Results

As several informal enterprises employ one or two persons and the vast majority do not employ more than four individuals (Section 3), the statistically significant effects attaching to the delicensing and FDI reforms discussed in Section 5.1 do not appear to be of economic relevance. For instance, the results indicate that in states with flexible labour markets, the average informal enterprise in an FDI liberalised industry grows by approximately 10 percent in employment terms. Assuming for convenience that the average informal enterprise in these states employs two persons, this translates to an increase of 0.2 persons; alternatively, on average and ceteris paribus, among every ten informal enterprises employing two persons, one is predicted to hire two additional individuals in response to FDI liberalisation. This suggests that the 'extensive margins' of industry-level employment and enterprise numbers may be a more meaningful consideration for the informal sector. In this section, I discuss results yielded by industry-level regressions for the informal manufacturing sector, following the discussion in Section 4.

Baseline results of this analysis are presented in Table 9, with the first three columns focusing on industry-level employment in informal enterprises as a whole and in OAMEs and NDMEs and the final three columns focusing on the number of enterprises for these three groups. As regards employment, I find that over the 1990–2001 period, delicensing is associated with a statistically significant increase of 41 percent on average informal industry size in states with inflexible labour markets, ceteris paribus. This increase is entirely attributable to increased industry-level employment in OAMEs (Panel C, Table 9). Furthermore, over the 1990–2001 period, FDI reform is associated with a significant average increase of approximately 55 percent in industry level OAME employment in states with flexible labour markets (Panel B, Table 9), ceteris paribus. In line with the enterprise-level difference results discussed in Section 5.1, no significance attaches to the reductions in final goods and input tariffs.

When I consider the implications of the reforms for informal enterprise numbers (Columns 4 to 6, Table 9), I find that input tariff declines are associated with significant increases in the number of OAMEs in a pan-Indian context (Panel A, Table 9). Further analysis suggests that these increases are restricted to states with inflexible labour markets, with significance weakening to the 10 percent level (Panel C, Table 9). Again, for the period in question (1990–2001), delicensing is associated with a significant increase of approximately 32 percent in OAME numbers in states with inflexible labour markets, and FDI liberalisation precedes a significant increment of 51.5 percent in OAME numbers in states with flexible labour markets.

These shifts in the number of OAMEs and industry-level employment in OAMEs associated with the delicensing and FDI reforms are both statistically significant and economically meaningful. In essence, controlling for industry and time fixed effects, delicensing and FDI liberalisation go hand-in-hand with a sizeable expansion of informal manufacturing industries in the 1990s. Interestingly, no significance attaches to any reform variable in the context of NDMEs in Table 9, both in terms of industry-level employment in these larger informal enterprises and in terms of their numbers. As both the delicensing and FDI reform instruments were targeted at relatively large formal firms (following the discussion in Section 2), it is probable that any informal sector impacts linked to these reforms arise on account of product market competition or collaborative linkages between formal firms and informal enterprises (Section 1). The data are better suited to an analysis of whether the observed effects are attributable to differences in the extent to which product markets are competitive. This motivates the following subsection.

**Table 9.** Economic reforms and informal sector employment: Industry-level difference effects (1990–2001).

| | ln (emp) | ln (emp) OAMEs | ln (emp) NDMEs | ln (ent) | ln (ent) OAMEs | ln (ent) NDMEs |
|---|---|---|---|---|---|---|
| **A: All states** | | | | | | |
| | (1) | (2) | (3) | | | |
| Final goods tariffs | 0.079 | 0.255 | −0.176 | 0.202 | 0.333 | −0.197 |
| | (0.239) | (0.329) | (0.319) | (0.255) | (0.292) | (0.326) |
| Input tariffs | −2.294 | −2.760 | 0.546 | −3.083 ** | −3.423 ** | 0.845 |
| | (1.412) | (1.685) | (1.276) | (1.475) | (1.615) | (1.401) |
| Delicensing | 0.274 ** | 0.290 ** | −0.018 | 0.184 | 0.188 | −0.042 |
| | (0.121) | (0.139) | (0.149) | (0.130) | (0.139) | (0.168) |
| FDI reform | 0.186 | 0.226 * | −0.088 | 0.178 | 0.199 | −0.108 |
| | (0.122) | (0.125) | (0.188) | (0.137) | (0.138) | (0.218) |
| Observations | 378 | 355 | 361 | 378 | 355 | 361 |
| R-squared | 0.120 | 0.133 | 0.069 | 0.112 | 0.117 | 0.073 |
| **B: States with flexible labour markets (FLEX 2 = 1)** | | | | | | |
| Final goods tariffs | 0.329 | 0.792 | −0.015 | 0.428 | 0.782 | −0.145 |
| | (0.393) | (0.505) | (0.452) | (0.416) | (0.480) | (0.359) |
| Input tariffs | −2.183 | −2.503 | −1.828 | −1.605 | −1.768 | −0.737 |
| | (1.850) | (2.442) | (1.944) | (1.893) | (2.275) | (1.802) |
| Delicensing | −0.030 | −0.051 | −0.072 | −0.064 | −0.073 | −0.059 |
| | (0.143) | (0.175) | (0.167) | (0.147) | (0.155) | (0.175) |
| FDI reform | 0.413 *** | 0.553 *** | 0.086 | 0.423 ** | 0.515 *** | 0.031 |
| | (0.137) | (0.157) | (0.188) | (0.169) | (0.184) | (0.197) |
| Observations | 327 | 302 | 315 | 327 | 302 | 315 |
| R-squared | 0.130 | 0.182 | 0.051 | 0.132 | 0.173 | 0.045 |
| **C: States with inflexible labour markets (FLEX 2 = 0)** | | | | | | |
| Final goods tariffs | 0.102 | 0.252 | −0.358 | 0.237 | 0.332 | −0.292 |
| | (0.359) | (0.487) | (0.340) | (0.351) | (0.416) | (0.408) |
| Input tariffs | −2.460 | −3.082 | 2.635 * | −3.946 * | −4.372 * | 2.418 |
| | (2.029) | (2.469) | (1.588) | (2.038) | (2.295) | (1.881) |
| Delicensing | 0.414 *** | 0.440 ** | −0.028 | 0.312 ** | 0.319 ** | −0.082 |
| | (0.155) | (0.177) | (0.169) | (0.147) | (0.159) | (0.197) |
| FDI reform | 0.150 | 0.179 | −0.186 | 0.142 | 0.150 | −0.170 |
| | (0.125) | (0.141) | (0.205) | (0.129) | (0.139) | (0.241) |
| Observations | 357 | 327 | 333 | 357 | 327 | 333 |
| R-squared | 0.137 | 0.159 | 0.043 | 0.147 | 0.156 | 0.039 |

Dependent variable: ln (emp) = natural logarithm of employment or ln (ent) = natural logarithm of number of enterprises. All regressions include a constant and industry and year fixed effects, and are weighted by pre-reform (1990) levels of the dependent variable. Standard errors, in brackets, are robust to heteroscedasticity. ***: Significant at 1% **: Significant at 5% *: Significant at 10%.

*5.3. Analysis of Mechanisms*

As discussed in Section 2, the reforms of the 1990s are likely to have engendered increases in product market competition in Indian manufacturing. While these shifts are of greater relevance for formal firms given the more direct effects of the reforms on the formal sector, there may have been spillovers into the informal sector. Several hypotheses may be proposed to describe the potential implications of competition between informal enterprises and formal firms. In one scenario, employment in formal firms in industries characterised by lower levels of pre-reform competition may have been less responsive to the reforms, with the competitive pressure exerted by new industry entrants being limited on account of incumbents enjoying substantial economies of scale and scope. Such industries

may arguably have been less affected by the reforms in a direct sense and may therefore have witnessed fewer spillovers into the informal sector.

An alternative view might posit that less competitive formal sector industries may have registered more, rather than fewer, shifts in employment in informal enterprises, with the informal sector functioning as a 'shock absorber' in the post-reform period. This hypothesis could be in consonance with the reforms engendering Melitz (2003)-type structural shifts within less competitive industries, with the least productive formal firms being forced to exit their markets. Moreover, any spillovers into the informal sector may have different implications for OAMEs and NDMEs, with the latter being larger informal units and therefore arguably more likely to compete with smaller formal firms as opposed to the purely household-based OAMEs.

One difficulty that arises in this context is a lack of data on competition between informal enterprises and formal firms. In 2001, the NSSO introduced a survey question regarding competition from larger firms being a problem that had been encountered by informal units, but no similar indicator is present in the survey data for 1990 and 1995. In light of this constraint, I hypothesise that competition between informal and formal market players is more likely to exist in industries characterised by a smaller gulf between informal and formal firm size. To test this hypothesis, I compute the ratio of average formal firm employment to average informal enterprise employment (the 'F-I ratio') in the pre-reform period (1990) for each three-digit industry. This ratio varies considerably across industries, with a median of 19.5 and a mean of 39.3. Intuitively, industries with a lower 'F-I' ratio are perhaps more likely to witness competition between formal and informal operators. Conversely, industries with a higher 'F-I' ratio might be expected to be less likely to be competitive in this sense, and perhaps more likely to feature 'collaboration' between the formal and informal sectors (in terms of supply chain linkages or agglomeration externalities).

Table 10 presents results separately for informal enterprises operating in three-digit industries characterised by relatively low and high pre-reform (1990) levels of this 'F-I' ratio, with Columns 2, 3 and 4 representing industries with a ratio below the first quartile, below the median and exceeding the median respectively. The results are striking: my baseline estimates are strengthened, in magnitude and significance, for industries with lower 'F-I' ratios, most visibly for industries where this ratio is below the first quartile (Column 2, Table 10). Conversely, for industries characterised by higher 'F-I' ratios, the baseline results lose significance (Column 4, Table 10). This might be viewed as evidence in favour of the reforms generating spillovers in the informal sector on account of competition between formal and informal operators.

In Table 11, I present industry-level findings corresponding to the industry groups analysed in Table 10. This yields a slightly more nuanced result: while the baseline results are in general more robust for more competitive industries (as characterised by lower 'F-I' ratios in 1990), the positive baseline association between FDI reform and informal enterprise numbers in states with flexible labour markets is restricted to industries with higher pre-reform 'F-I' ratios.

Furthermore, to explore the implications of competition within the formal sector for informal enterprise employment, I use the four-firm concentration ratio (CR4), which captures the proportion of each three-digit industry's output that is accounted for by the four largest firms in that industry. The lower the CR4 estimate, the more competitive an industry may be perceived to be, as the largest firms account for a relatively small share of industry output. Conversely, industries with higher CR4 ratios are arguably less competitive, with the largest firms commanding a more substantial market share. I use data from the Annual Survey of Industries (ASI), which focuses on formal firms in India, to compute the CR4 statistic for every three-digit industry in 1990, with output measured in terms of gross sale values. These data reveal that the CR4 estimate declined in most (90 percent of) formal sector industries in the 1990–1995 period, which is suggestive of increases in product market competition driven by the economic liberalisation of the 1990s.

**Table 10.** Economic reforms, labour market flexibility and employment in informal enterprises (1990–2001): Analysis based on the ratio of average formal firm employment to average informal enterprise employment ('F-I ratio') in 1990.

| | Baseline (All Enterprises) | Industry F-I Ratio below First Quartile in 1990 ('Most Competitive') | Industry F-I Ratio below Median in 1990 ('More Competitive') | Industry F-I Ratio above Median in 1990 ('Less Competitive') |
|---|---|---|---|---|
| Final goods tariffs | −0.132 | 0.014 | −0.013 | −0.335 |
| | (0.113) | (0.151) | (0.076) | (0.207) |
| Final goods tariffs * FLEX 2 | 0.151 | 0.055 | 0.005 | 0.387 ** |
| | (0.097) | (0.192) | (0.106) | (0.167) |
| Input tariffs | 0.576 * | 0.958 * | 0.524 | 0.798 |
| | (0.327) | (0.559) | (0.324) | (0.556) |
| Input tariffs * FLEX 2 | −0.178 | −0.189 | −0.050 | −0.359 |
| | (0.168) | (0.292) | (0.204) | (0.299) |
| Delicensing | 0.108 *** | 0.140 *** | 0.102 *** | 0.026 |
| | (0.030) | (0.043) | (0.030) | (0.065) |
| Delicensing * FLEX 2 | −0.088 ** | −0.114 * | −0.091 ** | −0.016 |
| | (0.037) | (0.064) | (0.046) | (0.055) |
| FDI reform | 0.024 | 0.005 | 0.014 | 0.044 |
| | (0.028) | (0.043) | (0.027) | (0.085) |
| FDI reform * FLEX 2 | 0.076 * | 0.174 *** | 0.099 ** | 0.042 |
| | (0.041) | (0.049) | (0.040) | (0.079) |
| **Flexible labour markets: Effects of changes in final goods tariffs** | | | | |
| (Final goods tariffs)*(1 + FLEX 2) | 0.019 | 0.069 | −0.008 | 0.053 |
| Standard Error | 0.078 | 0.160 | 0.081 | 0.126 |
| *p*-value (combined effect = 0) | 0.809 | 0.669 | 0.925 | 0.677 |
| **Flexible labour markets: Effects of changes in input tariffs** | | | | |
| (Input tariffs)*(1 + FLEX 2) | 0.398 | 0.769 | 0.474 | 0.439 |
| Standard Error | 0.300 | 0.573 | 0.319 | 0.574 |
| *p*-value (combined effect = 0) | 0.185 | 0.180 | 0.137 | 0.445 |
| **Flexible labour markets: Effects of delicensing** | | | | |
| (Delicensing)*(1 + FLEX 2) | 0.019 | 0.026 | 0.010 | 0.010 |
| Standard Error | 0.038 | 0.073 | 0.047 | 0.066 |
| *p*-value (combined effect = 0) | 0.613 | 0.721 | 0.827 | 0.885 |
| **Flexible labour markets: Effects of FDI reform** | | | | |
| (FDI reform)*(1 + FLEX 2) | 0.099 ** | 0.179 *** | 0.113 *** | 0.086 |
| Standard Error | 0.043 | 0.058 | 0.043 | 0.077 |
| *p*-value (combined effect = 0) | 0.021 | 0.002 | 0.009 | 0.265 |
| State FE | Yes | Yes | Yes | Yes |
| Year FE | Yes | Yes | Yes | Yes |
| Industry FE | Yes | Yes | Yes | Yes |
| Observations | 195,789 | 63,997 | 133,303 | 62,486 |
| R-squared | 0.164 | 0.091 | 0.115 | 0.199 |

Dependent variable: natural logarithm of total number of persons engaged; 'FE': fixed effects. Standard errors, in brackets, are clustered at the state-industry level. ***: Significant at 1% **: Significant at 5% *: Significant at 10%.

One hypothesis is that formal sector industries that had higher CR4 figures (and were therefore less competitive) prior to the reforms may have been more vulnerable to the increases in product market competition in the 1990s. As such, the informal sector may have expanded in these industries, following the exit of less productive formal firms from the market. This hypothesis is examined in Tables 12 and 13, which present enterprise and industry-level results for informal manufacturing in industries with higher and lower formal sector CR4 in 1990. At the enterprise level (Table 12), the delicensing effect holds for both groups of industries, while the FDI effect is robust only in industries featuring a higher degree of formal sector competition in 1990 (as captured by lower CR4 estimates). Conversely, at the industry level, all the baseline results concerning informal sector expansion in

response to the delicensing and FDI reforms hold only for industries with higher CR4 figures in 1990 (Table 13). This supports the view that economically meaningful structural shifts in the post-reform informal sector may have been driven by an evolving competitive landscape in the formal sector.

**Table 11.** Economic reforms and informal sector employment: Industry-level effects for enterprise numbers (1990–2001) based on the ratio of average formal firm employment to average informal enterprise employment ('F-I ratio') in 1990.

| | Dependent Variable: ln (Number of Informal Enterprises in Three-Digit Industry) | | |
|---|---|---|---|
| | **All Industries** | **Industries with F-I Ratio below Median in 1990 ('More Competitive')** | **Industries with F-I Ratio above Median in 1990 ('Less Competitive')** |
| **A: All states** | | | |
| Final goods tariffs | 0.202 | 0.292 | 0.554 |
| | (0.255) | (0.519) | (0.349) |
| Input tariffs | −3.083 ** | −4.730 * | 0.567 |
| | (1.475) | (2.407) | (2.466) |
| Delicensing | 0.184 | 0.253 * | 0.151 |
| | (0.130) | (0.134) | (0.319) |
| FDI reform | 0.178 | 0.128 | 0.326 |
| | (0.137) | (0.106) | (0.286) |
| Observations | 378 | 195 | 183 |
| R-squared | 0.112 | 0.326 | 0.029 |
| **B: States with flexible labour markets (FLEX 2 = 1)** | | | |
| Final goods tariffs | 0.428 | 0.346 | 1.302 * |
| | (0.416) | (0.516) | (0.665) |
| Input tariffs | −1.605 | −2.950 | −2.697 |
| | (1.893) | (2.461) | (3.691) |
| Delicensing | −0.064 | 0.053 | −0.399 |
| | (0.147) | (0.150) | (0.440) |
| FDI reform | 0.423 ** | 0.097 | 0.849 ** |
| | (0.169) | (0.190) | (0.322) |
| Observations | 327 | 168 | 159 |
| R-squared | 0.132 | 0.258 | 0.159 |
| **C: States with inflexible labour markets (FLEX 2 = 0)** | | | |
| Final goods tariffs | 0.237 | 0.880 | 0.160 |
| | (0.351) | (0.588) | (0.470) |
| Input tariffs | −3.946 * | −7.118 *** | 2.196 |
| | (2.038) | (2.627) | (3.917) |
| Delicensing | 0.312 ** | 0.279 * | 0.509 |
| | (0.147) | (0.146) | (0.357) |
| FDI reform | 0.142 | 0.201 * | −0.188 |
| | (0.129) | (0.115) | (0.443) |
| Observations | 357 | 198 | 159 |
| R-squared | 0.147 | 0.323 | 0.057 |

Dependent variable: natural logarithm of number of informal enterprises at the three-digit industry level. All regressions include a constant and industry and year fixed effects, and are weighted by pre-reform (1990) levels of the dependent variable. Standard errors, in brackets, are robust to heteroscedasticity. ***: Significant at 1%, **: Significant at 5%, *: Significant at 10%.

**Table 12.** Economic reforms, labour market flexibility and employment in informal enterprises (1990–2001): Analysis based on formal sector four firm concentration ratio (CR4) in 1990 (the proportion of formal industry-level output accounted for by the four largest formal firms in 1990).

| | Baseline (All Enterprises) | Enterprises in Industries with Formal CR4 above Median in 1990 (Less Competitive Formal Sector) | Enterprises in Industries with Formal CR4 below Median in 1990 (More Competitive Formal Sector) |
|---|---|---|---|
| Final goods tariffs | −0.132 | −0.200 | −0.109 |
| | (0.113) | (0.200) | (0.073) |
| Final goods tariffs * FLEX 2 | 0.151 | 0.165 | 0.167 * |
| | (0.097) | (0.184) | (0.087) |
| Input tariffs | 0.576 * | 0.460 | 0.184 |
| | (0.327) | (0.498) | (0.409) |
| Input tariffs * FLEX 2 | −0.178 | −0.246 | −0.121 |
| | (0.168) | (0.298) | (0.184) |
| Delicensing | 0.108 *** | 0.079 ** | 0.092 ** |
| | (0.030) | (0.036) | (0.043) |
| Delicensing * FLEX 2 | −0.088 ** | −0.108 ** | −0.018 |
| | (0.037) | (0.055) | (0.037) |
| FDI reform | 0.024 | 0.082 | 0.070 ** |
| | (0.028) | (0.140) | (0.032) |
| FDI reform * FLEX 2 | 0.076 * | −0.123 | 0.099 ** |
| | (0.041) | (0.144) | (0.042) |
| **Flexible labour markets: Effects of changes in final goods tariffs** | | | |
| **(Final goods tariffs)*(1 + FLEX 2)** | 0.019 | −0.036 | 0.058 |
| Standard Error | 0.078 | 0.148 | 0.070 |
| *p*-value (combined effect = 0) | 0.809 | 0.810 | 0.408 |
| **Flexible labour markets: Effects of changes in input tariffs** | | | |
| **(Input tariffs)*(1 + FLEX 2)** | 0.398 | 0.215 | 0.063 |
| Standard Error | 0.300 | 0.439 | 0.428 |
| *p*-value (combined effect = 0) | 0.185 | 0.625 | 0.883 |
| **Flexible labour markets: Effects of delicensing** | | | |
| **(Delicensing)*(1 + FLEX 2)** | 0.019 | −0.029 | 0.074 * |
| Standard Error | 0.038 | 0.056 | 0.040 |
| *p*-value (combined effect = 0) | 0.613 | 0.609 | 0.064 |
| **Flexible labour markets: Effects of FDI reform** | | | |
| **(FDI reform)*(1 + FLEX 2)** | 0.099 ** | −0.041 | 0.170 *** |
| Standard Error | 0.043 | 0.231 | 0.048 |
| *p*-value (combined effect = 0) | 0.021 | 0.859 | 0.000 |
| State FE | Yes | Yes | Yes |
| Year FE | Yes | Yes | Yes |
| Industry FE | Yes | Yes | Yes |
| Observations | 195,789 | 88,512 | 107,277 |
| R-squared | 0.164 | 0.122 | 0.206 |

Dependent variable: natural logarithm of number of paid employees; 'FE' denotes fixed effects. Standard errors, in brackets, are clustered at the state-industry level. ***: Significant at 1%, **: Significant at 5%, *: Significant at 10%.

**Table 13.** Economic reforms and informal sector employment: Industry-level effects for enterprise numbers (1990–2001) based on formal sector four firm concentration ratio (CR4) in 1990 (the proportion of formal industry-level output accounted for by the four largest formal firms in 1990).

| | Dependent Variable: ln (Number of Informal Enterprises in Three-Digit Industry) | | |
|---|---|---|---|
| | **All Industries** | **Industries with Formal CR4 above Median in 1990 (Less Competitive Formal Sector)** | **Industries with Formal CR4 below Median in 1990 (More Competitive Formal Sector)** |
| **A: All states** | | | |
| Final goods tariffs | 0.202 | 0.244 | 0.128 |
| | (0.255) | (0.198) | (0.429) |
| Input tariffs | −3.083** | −3.367 *** | 0.711 |
| | (1.475) | (0.809) | (1.829) |
| Delicensing | 0.184 | 0.405 *** | −0.201 |
| | (0.130) | (0.099) | (0.250) |
| FDI reform | 0.178 | 0.687* | −0.077 |
| | (0.137) | (0.356) | (0.227) |
| Observations | 378 | 153 | 225 |
| R-squared | 0.112 | 0.491 | 0.086 |
| **B: States with flexible labour markets (FLEX 2 = 1)** | | | |
| Final goods tariffs | 0.428 | 1.221 ** | −0.207 |
| | (0.416) | (0.480) | (0.503) |
| Input tariffs | −1.605 | −5.504 ** | 2.090 |
| | (1.893) | (2.441) | (1.420) |
| Delicensing | −0.064 | 0.215 * | −0.533 ** |
| | (0.147) | (0.121) | (0.252) |
| FDI reform | 0.423 ** | 0.284 *** | 0.303 |
| | (0.169) | (0.080) | (0.293) |
| Observations | 327 | 132 | 195 |
| R-squared | 0.132 | 0.463 | 0.119 |
| **C: States with inflexible labour markets (FLEX 2 = 0)** | | | |
| Final goods tariffs | 0.237 | −0.019 | 0.521 |
| | (0.351) | (0.169) | (0.578) |
| Input tariffs | −3.946 * | −2.908 *** | −0.056 |
| | (2.038) | (0.390) | (2.896) |
| Delicensing | 0.312 ** | 0.451 *** | 0.090 |
| | (0.147) | (0.140) | (0.200) |
| FDI reform | 0.142 | 0.433 | −0.103 |
| | (0.129) | (0.686) | (0.191) |
| Observations | 357 | 138 | 219 |
| R-squared | 0.147 | 0.482 | 0.121 |

Dependent variable: natural logarithm of number of informal enterprises at the three-digit industry level. All regressions include a constant and industry and year fixed effects, and are weighted by pre-reform (1990) levels of the dependent variable. Standard errors, in brackets, are robust to heteroscedasticity. ***: Significant at 1%, **: Significant at 5%, *: Significant at 10%.

While I use the CR4 as a baseline measure of intra-industry competition, I also consider two alternative metrics that are often used by market regulators in several economies around the globe: the eight-firm concentration ratio (CR8) and the Herfindahl-Hirschman Index (HHI).[12] The CR8 is

---

[12] See for instance https://www.justice.gov/atr/herfindahl-hirschman-index and http://eur-lex.europa.eu/legal-content/EN/ALL/?uri=CELEX:52004XC0205(02).

an extension of the CR4, and is defined as the proportion of each industry's output that is accounted for by the eight largest firms in that industry. The HHI is an alternative indicator of intra-industry competition that is obtained by aggregating the squares of the market shares of all the firms in an industry. It seeks to weight each firm in an industry in proportion to its output share in the industry. The findings discussed in this section are robust to using either the CR8 or the HHI instead of the CR4 (Section 5.4).

In summary, the evidence is broadly indicative of competition between formal and informal manufacturers, as also within the formal sector, being a mechanism underlying the implications of liberalisation, in particular the delicensing reform, for the informal sector. While the possibility of the results also deriving in part from collaborative linkages between informal and formal units cannot be ruled out, a rigorous examination of the same is beyond the scope of the current study and data.

*5.4. Further Analysis and Robustness Checks*

As explained in Section 2, the tariff declines that were phased in during the initial years of reform (1991–1997) were arguably an exogenous event, although tariff policy endogeneity might be an issue in the post-1997 period, when the pressure to adhere to externally imposed guidelines had waned. Although my dataset focuses on employment shifts in the 1990–2001 period and is therefore arguably largely immune to this concern, I explore whether tariff endogeneity poses problems for my results in several ways.

First, in Table A1 in Appendix A, I regress final goods and input tariffs on lagged industry-level employment (in logarithmic and absolute terms) and lagged industry employment shares for the informal sector in alternative specifications, including year and industry fixed effects throughout. The time lags used vary over one to three years. In all instances, there is no evidence of any association between informal industry employment levels and tariff rates in later years.

Next, I run separate regressions of the changes in final goods and input tariffs on the lagged changes in informal industry-level employment, including period and industry fixed effects throughout. As evidenced in Table A2 in Appendix A, there is no significant association between changes in informal employment and tariff changes in subsequent periods. Following Topalova and Khandelwal's 2011 formal sector analysis, I also confirm that the period-to-period final goods and input tariff changes are not correlated with pre-existing informal industry employment levels (Table A2 in Appendix A).

Furthermore, I drop two industries that were highly protected in the pre-reform period, yet were subjected to visibly low tariff declines relative to other industries with comparably high tariff rates in the 1991–1997 period. In Section 3, Figure 2a suggests that some endogeneity may have seeped into tariff policy as regards these two industries even in the face of the IMF backed reforms of 1991, given that the high degree of tariff protection enjoyed by these industries in the pre-reform period was relaxed to a lesser extent in the reform years relative to other industries with comparably high pre-reform tariffs. Column 2 of Table 14 reveals that the omission of these outliers leaves the baseline results (re-presented in Column 1 of Table 14 for convenience) virtually unchanged in terms of both magnitude and significance.

To assess whether my results are influenced by state-level characteristics other than the flexibility of labour market regulation, I run a regression in which I add state-year interaction fixed effects to my baseline specification. The results, presented in Column 3 of Table 14, indicate that the baseline results are similar in magnitude and significance following the addition of these interactions. This suggests that the baseline statistical significance of the interplay between the reforms and labour market flexibility is retained after accounting for other state-level trends. Interestingly, controlling for state-year trends, input tariff declines are associated with a significant decrease in informal enterprise-level employment, on average and ceteris paribus, in states with inflexible labour markets. As informal enterprises rarely use imported inputs, I interpret this as being a spillover effect driven by the possible general equilibrium price shifts engineered by reduced imported input prices (following the discussion

in Section 2). The concurrent currency devaluation means that such shifts are all but impossible to trace in contemporaneous price index data.

**Table 14.** Economic reforms, labour market flexibility and employment in informal enterprises (1990–2001): Tariff endogeneity check—Dropping outlier industries (wine manufacturing and the distillation, rectification and blending of spirits) and adding state-year interaction fixed effects.

| | Baseline (All Enterprises) | Dropping Outlier Industries | Adding State-Year Interactions |
|---|---|---|---|
| Final goods tariffs | −0.132 | −0.133 | −0.131 |
| | (0.113) | (0.114) | (0.106) |
| Final goods tariffs * FLEX 2 | 0.151 | 0.155 | 0.152 * |
| | (0.097) | (0.100) | (0.092) |
| Input tariffs | 0.576 * | 0.577* | 0.745 ** |
| | (0.327) | (0.328) | (0.310) |
| Input tariffs * FLEX 2 | −0.178 | −0.184 | −0.908 *** |
| | (0.168) | (0.171) | (0.298) |
| Delicensing | 0.108 *** | 0.108 *** | 0.092 *** |
| | (0.030) | (0.030) | (0.028) |
| Delicensing * FLEX 2 | −0.088 ** | −0.088 ** | −0.061 * |
| | (0.037) | (0.037) | (0.037) |
| FDI reform | 0.024 | 0.023 | 0.022 |
| | (0.028) | (0.028) | (0.027) |
| FDI reform * FLEX 2 | 0.076 * | 0.076 * | 0.081 ** |
| | (0.041) | (0.041) | (0.038) |
| **Flexible labour markets: Effects of changes in final goods tariffs** | | | |
| **(Final goods tariffs)*(1 + FLEX 2)** | 0.019 | 0.022 | 0.021 |
| Standard Error | 0.078 | 0.079 | 0.073 |
| *p*-value (combined effect = 0) | 0.809 | 0.785 | 0.770 |
| **Flexible labour markets: Effects of changes in input tariffs** | | | |
| **(Input tariffs)*(1 + FLEX 2)** | 0.398 | 0.393 | −0.163 |
| Standard Error | 0.300 | 0.301 | 0.328 |
| *p*-value (combined effect = 0) | 0.185 | 0.192 | 0.620 |
| **Flexible labour markets: Effects of delicensing** | | | |
| **(Delicensing)*(1 + FLEX 2)** | 0.019 | 0.019 | 0.031 |
| Standard Error | 0.038 | 0.038 | 0.036 |
| *p*-value (combined effect = 0) | 0.613 | 0.616 | 0.385 |
| **Flexible labour markets: Effects of FDI reform** | | | |
| **(FDI reform)*(1 + FLEX 2)** | 0.099 ** | 0.099 ** | 0.103 ** |
| Standard Error | 0.043 | 0.043 | 0.040 |
| *p*-value (combined effect = 0) | 0.021 | 0.021 | 0.010 |
| State FE | Yes | Yes | Yes |
| Year FE | Yes | Yes | Yes |
| Industry FE | Yes | Yes | Yes |
| State-Year FE | No | No | Yes |
| Observations | 195789 | 195724 | 195789 |
| R-squared | 0.164 | 0.164 | 0.169 |

Dependent variable: natural logarithm of total number of persons engaged; 'FE': fixed effects. Standard errors, in brackets, are clustered at the state-industry level. ***: Significant at 1%, **: Significant at 5%, *: Significant at 10%.

Column 2 of Table 15 attempts to control for the age, in terms of years of operation, of the informal enterprises in my dataset. The limitation of this attempt is that over 40 percent of the enterprises in the dataset do not provide estimates of the duration for which they have operated. Nonetheless, for enterprises providing these estimates, the baseline figures are strengthened in magnitude and

significance after controlling for enterprise age. Furthermore, the results also hold if the baseline measure of state-level labour market flexibility ('FLEX 2') is replaced by either the 'FLEX 1' measure or the 'FLEX 3' measure, both outlined in Section 3 (Column 3 and Column 4, Table 15).

**Table 15.** Economic reforms, labour market flexibility and employment in informal enterprises (1990–2001): Robustness checks—Accounting for enterprise age and alternative measures of labour market flexibility.

| | Baseline (All Enterprises) | Controlling for Enterprise Age (Where Reported) | FLEX 1 Instead of FLEX 2 | FLEX 3 Instead of FLEX 2 |
|---|---|---|---|---|
| Final goods tariffs | −0.132 | −0.259 *** | −0.091 | −0.058 |
| | (0.113) | (0.099) | (0.117) | (0.077) |
| Final goods tariffs * FLEX | 0.151 | 0.186 * | 0.030 | −0.088 |
| | (0.097) | (0.101) | (0.110) | (0.133) |
| Input tariffs | 0.576 * | 0.985 *** | 0.537 | 0.548 * |
| | (0.327) | (0.333) | (0.328) | (0.299) |
| Input tariffs * FLEX | −0.178 | −0.518 ** | 0.022 | 0.042 |
| | (0.168) | (0.251) | (0.187) | (0.206) |
| Delicensing | 0.108 *** | 0.126 *** | 0.084 *** | 0.088 *** |
| | (0.030) | (0.041) | (0.030) | (0.030) |
| Delicensing * FLEX | −0.088 ** | −0.107 ** | −0.027 | −0.029 |
| | (0.037) | (0.043) | (0.043) | (0.042) |
| FDI reform | 0.024 | 0.022 | 0.026 | 0.052 * |
| | (0.028) | (0.040) | (0.028) | (0.028) |
| FDI reform * FLEX | 0.076* | 0.095 ** | 0.077 * | −0.036 |
| | (0.041) | (0.045) | (0.041) | (0.038) |
| Enterprise age | | 0.003 *** | | |
| | | (0.001) | | |
| **Flexible labour markets: Effects of changes in final goods tariffs** | | | | |
| **(Final goods tariffs)*(1 + FLEX 2)** | 0.019 | −0.074 | −0.061 | −0.146 |
| Standard Error | 0.078 | 0.087 | 0.097 | 0.158 |
| *p*-value (combined effect = 0) | 0.809 | 0.397 | 0.529 | 0.357 |
| **Flexible labour markets: Effects of changes in input tariffs** | | | | |
| **(Input tariffs)*(1 + FLEX 2)** | 0.398 | 0.467 | 0.559 * | 0.590 |
| Standard Error | 0.300 | 0.349 | 0.316 | 0.364 |
| *p*-value (combined effect = 0) | 0.185 | 0.181 | 0.077 | 0.106 |
| **Flexible labour markets: Effects of delicensing** | | | | |
| **(Delicensing)*(1 + FLEX 2)** | 0.019 | 0.019 | 0.057 | 0.058 |
| Standard Error | 0.038 | 0.046 | 0.043 | 0.041 |
| *p*-value (combined effect = 0) | 0.613 | 0.682 | 0.181 | 0.151 |
| **Flexible labour markets: Effects of FDI reform** | | | | |
| **(FDI reform)*(1 + FLEX 2)** | 0.099 ** | 0.117 ** | 0.103 ** | 0.016 |
| Standard Error | 0.043 | 0.047 | 0.041 | 0.039 |
| *p*-value (combined effect = 0) | 0.021 | 0.012 | 0.013 | 0.690 |
| State FE | Yes | Yes | Yes | Yes |
| Year FE | Yes | Yes | Yes | Yes |
| Industry FE | Yes | Yes | Yes | Yes |
| Observations | 195,789 | 115,543 | 195,789 | 195,789 |
| R-squared | 0.164 | 0.160 | 0.163 | 0.163 |

Dependent variable: natural logarithm of total number of persons engaged; 'FE' denotes fixed effects. Standard errors, in brackets, are clustered at the state-industry level. The measure of labour market flexibility used in Columns 1 and 2 is the 'FLEX 2' measure, while Columns 3 and 4 use alternative measures, as specified in the column headings. ***: Significant at 1%, **: Significant at 5%, *: Significant at 10%.

In the baseline results discussed in Section 5.1, as well as in the findings presented in this section, the reform measures have been lagged by two years. In Table 16, I examine the extent to which the baseline figures in Column 6 of Table 5 are affected if a one-year or three-year lag is used instead of a

two-year lag. Table 16 suggests that these modifications yield figures that are similar in magnitude and significance to the baseline numbers. The exception is the employment enhancing effect associated with FDI reform in states with flexible labour markets, which loses significance when a one-year lag is used, although it is significant when a three-year lag is used. This suggests that the effect of FDI reform is likely to be a relatively slower, longer-lasting impact. Conversely, the weak baseline significance attaching to input tariff declines is strengthened if the time lag is reduced to one year, but disappears for a three-year lag, which is suggestive of a less lasting effect.

**Table 16.** Economic reforms, labour market flexibility and employment in informal enterprises (1990–2001): Robustness checks—Modifying the baseline reform time lag and excluding individual post-reform cross-sections.

| | Baseline (All Enterprises) | Time Lag: 1 Year | Time Lag: 3 Years |
|---|---|---|---|
| Final goods tariffs | −0.132 | −0.114 | −0.179 |
| | (0.113) | (0.101) | (0.110) |
| Final goods tariffs * FLEX 2 | 0.151 | 0.082 | 0.151 |
| | (0.097) | (0.098) | (0.093) |
| Input tariffs | 0.576 * | 0.850 *** | 0.543 |
| | (0.327) | (0.276) | (0.337) |
| Input tariffs * FLEX 2 | −0.178 | −0.087 | −0.173 |
| | (0.168) | (0.220) | (0.154) |
| Delicensing | 0.108 *** | 0.091 *** | 0.109 *** |
| | (0.030) | (0.028) | (0.029) |
| Delicensing * FLEX 2 | −0.088 ** | −0.078 ** | −0.092 ** |
| | (0.037) | (0.038) | (0.037) |
| FDI reform | 0.024 | 0.035 | 0.019 |
| | (0.028) | (0.027) | (0.028) |
| FDI reform * FLEX 2 | 0.076 * | 0.007 | 0.075 * |
| | (0.041) | (0.040) | (0.040) |
| **Flexible labour markets: Effects of changes in final goods tariffs** | | | |
| **(Final goods tariffs)*(1 + FLEX 2)** | 0.019 | −0.032 | −0.028 |
| Standard Error | 0.078 | 0.083 | 0.075 |
| *p*-value (combined effect = 0) | 0.809 | 0.704 | 0.710 |
| **Flexible labour markets: Effects of changes in input tariffs** | | | |
| **(Input tariffs)*(1 + FLEX 2)** | 0.398 | 0.763 *** | 0.370 |
| Standard Error | 0.300 | 0.292 | 0.318 |
| *p*-value (combined effect = 0) | 0.185 | 0.009 | 0.244 |
| **Flexible labour markets: Effects of delicensing** | | | |
| **(Delicensing)*(1 + FLEX 2)** | 0.019 | 0.014 | 0.017 |
| Standard Error | 0.038 | 0.037 | 0.037 |
| *p*-value (combined effect = 0) | 0.613 | 0.715 | 0.657 |
| **Flexible labour markets: Effects of FDI reform** | | | |
| **(FDI reform)*(1 + FLEX 2)** | 0.099 ** | 0.042 | 0.094 ** |
| Standard Error | 0.043 | 0.042 | 0.042 |
| *p*-value (combined effect = 0) | 0.021 | 0.317 | 0.027 |
| State FE | Yes | Yes | Yes |
| Year FE | Yes | Yes | Yes |
| Industry FE | Yes | Yes | Yes |
| Observations | 195,789 | 195,789 | 195,789 |
| R-squared | 0.164 | 0.165 | 0.164 |

Dependent variable: natural logarithm of total number of persons engaged; 'FE': fixed effects. Standard errors, in brackets, are clustered at the state-industry level. ***: Significant at 1%, **: Significant at 5%, *: Significant at 10%.

The results of the supplementary checks outlined in Section 3 are presented in Appendix A. Column 2 of Table A3 shows that retaining DMEs, the large informal enterprises that are excluded from the main analysis as they do not appear in the 1990 data (Section 3), has little impact on the baseline

results. Column 3 of Table A3 indicates that changing the 'FLEX 2' indicator value for Delhi and Jammu & Kashmir from 0 to 1 does not affect the key findings. Furthermore, Column 4 of Table A3 highlights that the baseline findings are virtually unchanged if the small proportion of informal enterprises employing ten or more individuals, omitted in the main analysis, are retained. Column 5 of Table A3 confirms that the inclusion of informal enterprises reporting zero or missing values for raw material usage and/or physical product manufacturing has no material impact on the baseline, while Column 6 of Table A3 suggests that this also applies to the use of input tariffs deriving only from final goods tariffs for manufacturing industries. In addition, following Aghion et al. 2008, I explore whether dropping individual states from my regressions has an impact on my results and conclude that the baseline numbers are robust to this check (Tables A4–A6).

Finally, following the discussion in Section 5.3, I examine the robustness of the findings regarding variations in intra-industry product market competition to the use of two alternative metrics of this competition, the CR8 and the HHI (outlined in Section 5.3). Tables A7 and A8 show that the results yielded by the use of these metrics are very similar to those deriving from the use of the CR4 (presented in Tables 12 and 13).

## 6. Discussion and Conclusions

This paper exploits the initiation of a quasi-exogenous round of tariff liberalisation and concurrent domestic policy reform to examine employment changes in small, unregistered (informal) Indian manufacturing enterprises in the 1990s. It also analyses the extent to which differences in state-level labour market flexibility influence these changes. To the best of my knowledge, this is the first study that focuses on informal manufacturers in this context, which is vital given that these firms account for the lion's share of employment in Indian manufacturing.

The results point to India's delicensing and FDI reforms being associated with significant shifts in informal sector employment. On average and ceteris paribus, delicensing (FDI reform) is associated with a statistically significant increase (increase) in employment in informal enterprises in states with inflexible (flexible) labour markets. More importantly, at a broader industry level, delicensing (FDI) reform is also a predictor of significant and, from an economic perspective, more meaningful increases in informal enterprise numbers in states with inflexible (flexible) labour markets. Furthermore, the swingeing import tariff reductions undertaken in India as part of the reform initiative of the 1990s rarely drive significant changes in informal sector employment, which is perhaps attributable to the fact that informal enterprises rarely engage in international trade.

Further analysis on the mechanisms driving these effects is suggestive of the implications of delicensing being considerably more prominent in industries with a higher propensity towards competing formal and informal operators, as also greater degrees of intra-formal sector competition. For FDI liberalisation, enterprise-level findings also hold particularly for these two sets of 'more competitive' industries, but industry-level shifts appear to be restricted to industries characterised by less competition in both instances. These findings are indicative of competition more clearly being a mechanism underpinning the impact of delicensing as a driver of formal and informal sector employment shifts, visible in states with inflexible labour markets. As regards FDI reform, there is more room to speculate that, while the degree of competition within the formal sector matters, supply chain linkages or agglomeration-based 'collaboration' between formal and informal players might also have a bearing on employment effects, primarily in states with flexible labour markets. This may be a fertile avenue for future research, particularly in instances where more refined data on FDI are available.

As policy makers in developing economies tend to emphasise increases in formal employment as a key goal of economic liberalisation, the findings of this paper are of general interest. They contribute to the growing literature examining the role of interactions between the Indian reform programme and variations in domestic state-level institutional characteristics in driving post-reform economic outcomes. The results highlight that an analysis of the implications of market reform for firm-level

employment is incomplete unless variations in regional labour market flexibility are accounted for. In a developing country setting characterised by a substantial informal sector, my findings strongly suggest that informal enterprises merit at least as much analysis as the formal sector. Data permitting, further research is eminently desirable, in particular on the linkages between the formal and informal sectors and the mechanisms underlying the impacts analysed in this study.

**Funding:** Funding received from the Department of Economics of the University of Sussex for purchasing survey data from the Government of India is gratefully acknowledged.

**Acknowledgments:** The author gratefully acknowledges feedback received from Dimitra Petropoulou, Andrew Newell and Amalavoyal Chari in particular, and more generally from faculty and research students at the University of Sussex. In addition, the author is grateful to Shanthi Nataraj of the RAND Corporation for sharing her data on India's manufacturing import tariff rates, delicensing and FDI reforms, and for providing guidance and clarification on several issues of relevance to this study. The author also appreciates constructive feedback offered by a large number of participants at the DEGIT XX Conference 2015 in Geneva, the European Trade Study Group (ETSG) Conference 2015 in Paris, the Royal Economic Society (RES) Conference 2015 in Manchester, and the 14th Global Economic Policy (GEP) Annual Postgraduate Conference 2015 in Nottingham.

**Conflicts of Interest:** The author declares no conflict of interest.

## Appendix A

**Table A1.** Tariff endogeneity check—regression of tariffs on lagged informal industry employment.

| Period (Dependent Variable) | t + 1 | t + 2 | t + 3 |
| --- | --- | --- | --- |
| **Dependent variable: Final goods tariffs** | | | |
| ln (Informal employment) | −0.016 | −0.005 | 0.012 |
| | (0.032) | (0.006) | (0.013) |
| Absolute informal employment | −0.000 | −0.000 * | 0.000 |
| | (0.000) | (0.000) | (0.000) |
| Share of informal employment | −1.357 * | −0.149 * | 0.266 |
| | (0.619) | (0.073) | (0.200) |
| **Dependent variable: Input tariffs** | | | |
| ln (Informal employment) | −0.012 | 0.007 | 0.004 |
| | (0.009) | (0.010) | (0.009) |
| Absolute informal employment | −0.000 *** | 0.000 | 0.000 |
| | (0.000) | (0.000) | (0.000) |
| Share of informal employment | −0.430 *** | 0.351 * | 0.250 |
| | (0.080) | (0.148) | (0.133) |

The independent variables are measured in period t. All specifications include year and industry fixed effects. Standard errors, in parentheses, are robust to heteroscedasticity. ***: Significant at 1%, *: Significant at 10%.

**Table A2.** Tariff endogeneity check—regression of changes in tariffs on lagged changes in informal employment (industry level).

| Period (Dependent Variable) | t + 1 | t + 2 | t + 3 |
|---|---|---|---|
| **Dependent variable: Change in final goods tariffs** | | | |
| Change in ln (informal employment) | −0.046 | −0.017 | 0.002 |
| | (0.030) | (0.010) | (0.009) |
| ln (informal employment) | −0.057 | −0.018306 | 0.003901 |
| | (0.048) | (0.017325) | (0.014173) |
| Change in absolute informal employment | −0.000 | −0.000 | 0.000 |
| | (0.000) | (0.000) | (0.000) |
| Absolute informal employment | −0.000 | −0.000 | 0.000 |
| | (0.000) | (0.000) | (0.000) |
| **Dependent variable: Change in input tariffs** | | | |
| Change in ln (informal employment) | −0.021 * | 0.007 | 0.007 |
| | (0.009) | (0.011) | (0.008) |
| ln (informal employment) | −0.034 * | 0.007 | 0.008 |
| | (0.013) | (0.019) | (0.015) |
| Change in absolute informal employment | −0.000 *** | 0.000 | 0.000 |
| | (0.000) | (0.000) | (0.000) |
| Absolute informal employment | −0.000 *** | 0.000 | 0.000 |
| | (0.000) | (0.000) | (0.000) |

The independent variables are measured in period t. All specifications include period and industry fixed effects. Standard errors, in parentheses, are clustered at the state-industry level. ***: Significant at 1%, *: Significant at 10%.

**Table A3.** Economic reforms, labour market flexibility and employment in informal enterprises (1990–2001): Additional robustness checks.

| | Baseline | With DMEs for 1995 and 2001 | Change in 'FLEX 2' Value for Delhi and Jammu & Kashmir (0 to 1) | Including Enterprises with Ten or More Persons Engaged | Including Enterprises Reporting Zero or No Value for Raw Material Use/Physical Products | Using Input Tariffs Based on Final Goods Tariffs for Manufacturing Industries Only |
|---|---|---|---|---|---|---|
| Final goods tariffs | −0.132 | −0.128 | −0.137 | −0.124 | −0.172* | −0.119 |
| | (0.113) | (0.106) | (0.115) | (0.112) | (0.090) | (0.126) |
| Final goods tariffs * FLEX 2 | 0.151 | 0.174 * | 0.156 | 0.151 | 0.158 | 0.111 |
| | (0.097) | (0.104) | (0.098) | (0.097) | (0.098) | (0.109) |
| Input tariffs | 0.576 * | 0.185 | 0.583 * | 0.558 * | 0.517 * | 0.444 |
| | (0.327) | (0.349) | (0.327) | (0.329) | (0.278) | (0.339) |
| Input tariffs * FLEX 2 | −0.178 | −0.222 | −0.179 | −0.161 | −0.139 | −0.051 |
| | (0.168) | (0.195) | (0.165) | (0.168) | (0.172) | (0.119) |
| Delicensing | 0.108 *** | 0.113 *** | 0.105 *** | 0.108 *** | 0.057 ** | 0.095 *** |
| | (0.030) | (0.031) | (0.030) | (0.030) | (0.027) | (0.028) |
| Delicensing * FLEX 2 | −0.088 ** | −0.061 | −0.079 ** | −0.081 ** | −0.047 | −0.081 ** |
| | (0.037) | (0.042) | (0.037) | (0.038) | (0.037) | (0.038) |
| FDI reform | 0.024 | 0.053 * | 0.022 | 0.025 | −0.000 | 0.029 |
| | (0.028) | (0.029) | (0.028) | (0.028) | (0.022) | (0.029) |
| FDI reform * FLEX 2 | 0.076 * | 0.160 *** | 0.079 ** | 0.074* | 0.070 ** | 0.077 * |
| | (0.041) | (0.048) | (0.040) | (0.041) | (0.034) | (0.041) |
| **Flexible labour markets: Effects of changes in final goods tariffs** | | | | | | |
| **(Final goods tariffs)*(1 + FLEX 2)** | 0.019 | 0.046 | 0.018 | 0.028 | −0.014 | −0.008 |
| Standard Error | 0.078 | 0.085 | 0.077 | 0.078 | 0.075 | 0.084 |
| *p*-value (combined effect = 0) | 0.809 | 0.589 | 0.810 | 0.721 | 0.853 | 0.924 |
| **Flexible labour markets: Effects of changes in input tariffs** | | | | | | |
| **(Input tariffs)*(1 + FLEX 2)** | 0.398 | −0.037 | 0.404 | 0.397 | 0.378 | 0.393 |
| Standard Error | 0.300 | 0.338 | 0.297 | 0.304 | 0.249 | 0.299 |
| *p*-value (combined effect = 0) | 0.185 | 0.914 | 0.173 | 0.191 | 0.128 | 0.188 |
| **Flexible labour markets: Effects of delicensing** | | | | | | |
| **(Delicensing)*(1 + FLEX 2)** | 0.019 | 0.052 | 0.026 | 0.027 | 0.010 | 0.014 |
| Standard Error | 0.038 | 0.041 | 0.037 | 0.038 | 0.035 | 0.039 |
| *p*-value (combined effect = 0) | 0.613 | 0.207 | 0.487 | 0.480 | 0.773 | 0.709 |

<div align="center">

**Table A3.** *Cont.*

</div>

| | Baseline | With DMEs for 1995 and 2001 | Change in 'FLEX 2' Value for Delhi and Jammu & Kashmir (0 to 1) | Including Enterprises with Ten or More Persons Engaged | Including Enterprises Reporting Zero or No Value for Raw Material Use/Physical Products | Using Input Tariffs Based on Final Goods Tariffs for Manufacturing Industries Only |
|---|---|---|---|---|---|---|
| | | | **Flexible labour markets: Effects of FDI reform** | | | |
| **(FDI reform)*(1 + FLEX 2)** | 0.099 ** | 0.213 *** | 0.101 ** | 0.099 ** | 0.070 ** | 0.106 |
| Standard Error | 0.043 | 0.051 | 0.042 | 0.043 | 0.030 | 0.043 |
| *p*-value (combined effect = 0) | 0.021 | 0.000 | 0.016 | 0.022 | 0.018 | 0.014 |
| State FE | Yes | Yes | Yes | Yes | Yes | Yes |
| Year FE | Yes | Yes | Yes | Yes | Yes | Yes |
| Industry FE | Yes | Yes | Yes | Yes | Yes | Yes |
| Observations | 195,789 | 216,456 | 195,789 | 196,001 | 316,755 | 195,789 |
| R-squared | 0.164 | 0.189 | 0.164 | 0.163 | 0.174 | 0.164 |

Dependent variable: natural logarithm of total number of persons engaged; 'FE' denotes fixed effects. Standard errors, in brackets, are clustered at the state-industry level. ***: Significant at 1%, **: Significant at 5%, *: Significant at 10%.

**Table A4.** Economic reforms, labour market flexibility and employment in informal enterprises (1990–2001): Excluding individual states (I).

| *Excluding:* | **Andhra Pradesh** | **Assam** | **Bihar** | **Gujarat** | **Haryana** | **Karnataka** |
|---|---|---|---|---|---|---|
| Final goods tariffs | −0.148 | −0.128 | −0.156 | −0.122 | −0.136 | −0.127 |
| | (0.116) | (0.116) | (0.125) | (0.116) | (0.114) | (0.116) |
| Final goods tariffs * FLEX 2 | 0.201 ** | 0.134 | 0.167 | 0.117 | 0.159 | 0.169* |
| | (0.101) | (0.098) | (0.105) | (0.100) | (0.098) | (0.098) |
| Input tariffs | 0.652 * | 0.559 * | 0.589 * | 0.557 * | 0.578 * | 0.473 |
| | (0.338) | (0.337) | (0.350) | (0.336) | (0.330) | (0.328) |
| Input tariffs * FLEX 2 | −0.182 | −0.143 | −0.247 | −0.121 | −0.187 | −0.209 |
| | (0.185) | (0.170) | (0.174) | (0.173) | (0.168) | (0.174) |
| Delicensing | 0.099 *** | 0.106 *** | 0.109 *** | 0.105 *** | 0.107 *** | 0.109 *** |
| | (0.030) | (0.030) | (0.031) | (0.030) | (0.030) | (0.030) |
| Delicensing * FLEX 2 | −0.059 | −0.086 ** | −0.098 ** | −0.086 ** | −0.088 ** | −0.105 *** |
| | (0.036) | (0.038) | (0.038) | (0.039) | (0.038) | (0.037) |
| FDI reform | 0.012 | 0.027 | 0.033 | 0.024 | 0.025 | 0.028 |
| | (0.029) | (0.028) | (0.030) | (0.028) | (0.028) | (0.028) |
| FDI reform * FLEX 2 | 0.062 | 0.068 * | 0.070 * | 0.063 | 0.074 * | 0.071 * |
| | (0.045) | (0.041) | (0.042) | (0.043) | (0.041) | (0.043) |
| **Flexible labour markets: Effects of changes in final goods tariffs** | | | | | | |
| **(Final goods tariffs)*(1 + FLEX 2)** | 0.054 | 0.006 | 0.011 | −0.005 | 0.023 | 0.042 |
| Standard Error | 0.084 | 0.079 | 0.079 | 0.082 | 0.079 | 0.079 |
| *p*-value (combined effect = 0) | 0.524 | 0.943 | 0.889 | 0.954 | 0.769 | 0.598 |
| **Flexible labour markets: Effects of changes in input tariffs** | | | | | | |
| **(Input tariffs)*(1 + FLEX 2)** | 0.470 | 0.416 | 0.342 | 0.436 | 0.391 | 0.264 |
| Standard Error | 0.303 | 0.306 | 0.313 | 0.310 | 0.302 | 0.306 |
| *p*-value (combined effect = 0) | 0.122 | 0.174 | 0.275 | 0.161 | 0.196 | 0.390 |
| **Flexible labour markets: Effects of delicensing** | | | | | | |
| **(Delicensing)*(1 + FLEX 2)** | 0.040 | 0.020 | 0.010 | 0.019 | 0.020 | 0.004 |
| Standard Error | 0.040 | 0.038 | 0.038 | 0.040 | 0.038 | 0.037 |
| *p*-value (combined effect = 0) | 0.314 | 0.600 | 0.790 | 0.635 | 0.604 | 0.921 |
| **Flexible labour markets: Effects of FDI reform** | | | | | | |
| **(FDI reform)*(1 + FLEX 2)** | 0.074 | 0.095 ** | 0.103 ** | 0.088 * | 0.099 ** | 0.099 ** |
| Standard Error | 0.048 | 0.043 | 0.044 | 0.045 | 0.043 | 0.045 |
| *p*-value (combined effect = 0) | 0.120 | 0.027 | 0.018 | 0.054 | 0.022 | 0.026 |
| State FE | Yes | Yes | Yes | Yes | Yes | Yes |
| Year FE | Yes | Yes | Yes | Yes | Yes | Yes |
| Industry FE | Yes | Yes | Yes | Yes | Yes | Yes |
| Observations | 175,093 | 190,029 | 177,854 | 188,617 | 192,790 | 188,039 |
| R-squared | 0.165 | 0.167 | 0.163 | 0.160 | 0.164 | 0.161 |

Dependent variable: natural logarithm of total number of persons engaged; 'FE' denotes fixed effects. Standard errors, in brackets, are clustered at the state-industry level. ***: Significant at 1%, **: Significant at 5%, *: Significant at 10%.

**Table A5.** Economic reforms, labour market flexibility and employment in informal enterprises (1990–2001): Excluding individual states (II).

| *Excluding:* | Kerala | Madhya Pradesh | Maharashtra | Orissa | Punjab | Rajasthan |
|---|---|---|---|---|---|---|
| Final goods tariffs | −0.140 | −0.151 | −0.119 | −0.132 | −0.135 | −0.120 |
| | (0.118) | (0.115) | (0.117) | (0.119) | (0.117) | (0.116) |
| Final goods tariffs * FLEX 2 | 0.162 | 0.177 * | 0.114 | 0.156 | 0.155 | 0.150 |
| | (0.100) | (0.098) | (0.107) | (0.099) | (0.099) | (0.100) |
| Input tariffs | 0.541 | 0.556 * | 0.611 * | 0.438 | 0.569 * | 0.565 * |
| | (0.336) | (0.330) | (0.341) | (0.344) | (0.334) | (0.334) |
| Input tariffs * FLEX 2 | −0.164 | −0.229 | −0.158 | −0.152 | −0.180 | −0.165 |
| | (0.172) | (0.167) | (0.183) | (0.177) | (0.170) | (0.172) |
| Delicensing | 0.108 *** | 0.113 *** | 0.106 *** | 0.114 *** | 0.107 *** | 0.106 *** |
| | (0.030) | (0.030) | (0.030) | (0.034) | (0.030) | (0.030) |
| Delicensing * FLEX 2 | −0.081 ** | −0.093 ** | −0.083 ** | −0.082 ** | −0.088 ** | −0.080 ** |
| | (0.038) | (0.038) | (0.042) | (0.039) | (0.038) | (0.040) |
| FDI reform | 0.023 | 0.022 | 0.024 | 0.032 | 0.023 | 0.022 |
| | (0.029) | (0.030) | (0.028) | (0.030) | (0.029) | (0.029) |
| FDI reform * FLEX 2 | 0.086 ** | 0.080 * | 0.102 *** | 0.087 ** | 0.078 * | 0.081 * |
| | (0.041) | (0.041) | (0.039) | (0.040) | (0.041) | (0.043) |
| **Flexible labour markets: Effects of changes in final goods tariffs** | | | | | | |
| **(Final goods tariffs)*(1 + FLEX 2)** | 0.022 | 0.026 | −0.005 | 0.024 | 0.020 | 0.030 |
| Standard Error | 0.079 | 0.079 | 0.095 | 0.080 | 0.079 | 0.081 |
| *p*-value (combined effect = 0) | 0.779 | 0.744 | 0.954 | 0.769 | 0.801 | 0.717 |
| **Flexible labour markets: Effects of changes in input tariffs** | | | | | | |
| **(Input tariffs)*(1 + FLEX 2)** | 0.377 | 0.327 | 0.453 | 0.286 | 0.388 | 0.400 |
| Standard Error | 0.305 | 0.305 | 0.322 | 0.316 | 0.304 | 0.308 |
| *p*-value (combined effect = 0) | 0.217 | 0.283 | 0.160 | 0.366 | 0.202 | 0.194 |
| **Flexible labour markets: Effects of delicensing** | | | | | | |
| **(Delicensing)*(1 + FLEX 2)** | 0.027 | 0.020 | 0.024 | 0.032 | 0.020 | 0.026 |
| Standard Error | 0.038 | 0.039 | 0.043 | 0.039 | 0.038 | 0.041 |
| *p*-value (combined effect = 0) | 0.487 | 0.612 | 0.586 | 0.411 | 0.608 | 0.528 |
| **Flexible labour markets: Effects of FDI reform** | | | | | | |
| **(FDI reform)*(1 + FLEX 2)** | 0.109 ** | 0.102 ** | 0.126 *** | 0.119 *** | 0.100 ** | 0.103 ** |
| Standard Error | 0.043 | 0.043 | 0.041 | 0.043 | 0.043 | 0.046 |
| *p*-value (combined effect = 0) | 0.012 | 0.017 | 0.002 | 0.006 | 0.020 | 0.024 |
| State FE | Yes | Yes | Yes | Yes | Yes | Yes |
| Year FE | Yes | Yes | Yes | Yes | Yes | Yes |
| Industry FE | Yes | Yes | Yes | Yes | Yes | Yes |
| Observations | 187,903 | 184,167 | 181,312 | 184,632 | 189,682 | 187,722 |
| R-squared | 0.159 | 0.166 | 0.164 | 0.167 | 0.163 | 0.165 |

Dependent variable: natural logarithm of total number of persons engaged; 'FE' denotes fixed effects. Standard errors, in brackets, are clustered at the state-industry level. ***: Significant at 1%, **: Significant at 5%, *: Significant at 10%.

**Table A6.** Economic reforms, labour market flexibility and employment in informal enterprises (1990–2001): Excluding individual states (III).

| *Excluding:* | **Tamil Nadu** | **Uttar Pradesh** | **West Bengal** | **Jammu & Kashmir** | **Delhi** |
|---|---|---|---|---|---|
| Final goods tariffs | −0.141 | −0.025 | −0.152 | −0.135 | −0.134 |
| | (0.113) | (0.071) | (0.131) | (0.115) | (0.114) |
| Final goods tariffs * FLEX 2 | 0.140 | 0.106 | 0.106 | 0.157 | 0.150 |
| | (0.098) | (0.087) | (0.108) | (0.098) | (0.098) |
| Input tariffs | 0.838 ** | 0.331 | 0.654 * | 0.588 * | 0.572 * |
| | (0.327) | (0.304) | (0.362) | (0.328) | (0.330) |
| Input tariffs * FLEX 2 | −0.220 | −0.135 | −0.117 | −0.194 | −0.171 |
| | (0.164) | (0.169) | (0.180) | (0.168) | (0.169) |
| Delicensing | 0.109 *** | 0.098 *** | 0.123 *** | 0.104 *** | 0.107 *** |
| | (0.030) | (0.031) | (0.029) | (0.030) | (0.030) |
| Delicensing * FLEX 2 | −0.121 *** | −0.079 ** | −0.100 *** | −0.086 ** | −0.087 ** |
| | (0.040) | (0.038) | (0.038) | (0.038) | (0.037) |
| FDI reform | 0.020 | 0.034 | −0.022 | 0.021 | 0.024 |
| | (0.029) | (0.029) | (0.032) | (0.028) | (0.028) |
| FDI reform * FLEX 2 | 0.092 ** | 0.050 | 0.088 ** | 0.084 ** | 0.076 * |
| | (0.045) | (0.041) | (0.041) | (0.041) | (0.041) |
| **Flexible labour markets: Effects of changes in final goods tariffs** | | | | | |
| **(Final goods tariffs)*(1 + FLEX 2)** | −0.001 | 0.081 | −0.045 | 0.022 | 0.016 |
| Standard Error | 0.085 | 0.069 | 0.079 | 0.079 | 0.079 |
| *p*-value (combined effect = 0) | 0.990 | 0.245 | 0.568 | 0.779 | 0.836 |
| **Flexible labour markets: Effects of changes in input tariffs** | | | | | |
| **(Input tariffs)*(1 + FLEX 2)** | 0.618 ** | 0.196 | 0.536 | 0.394 | 0.401 |
| Standard Error | 0.316 | 0.291 | 0.329 | 0.301 | 0.302 |
| *p*-value (combined effect = 0) | 0.050 | 0.500 | 0.103 | 0.190 | 0.185 |
| **Flexible labour markets: Effects of delicensing** | | | | | |
| **(Delicensing)*(1 + FLEX 2)** | −0.013 | 0.019 | 0.023 | 0.017 | 0.020 |
| Standard Error | 0.038 | 0.038 | 0.039 | 0.038 | 0.038 |
| *p*-value (combined effect = 0) | 0.733 | 0.621 | 0.558 | 0.648 | 0.598 |
| **Flexible labour markets: Effects of FDI reform** | | | | | |
| **(FDI reform)*(1 + FLEX 2)** | 0.112 ** | 0.084 * | 0.066 | 0.106 ** | 0.100 ** |
| Standard Error | 0.048 | 0.043 | 0.042 | 0.043 | 0.043 |
| *p*-value (combined effect = 0) | 0.019 | 0.051 | 0.110 | 0.015 | 0.020 |
| State FE | Yes | Yes | Yes | Yes | Yes |
| Year FE | Yes | Yes | Yes | Yes | Yes |
| Industry FE | Yes | Yes | Yes | Yes | Yes |
| Observations | 173,377 | 173,033 | 171,913 | 192,443 | 194,018 |
| R-squared | 0.163 | 0.168 | 0.181 | 0.165 | 0.164 |

Dependent variable: natural logarithm of total number of persons engaged; 'FE' denotes fixed effects. Standard errors, in brackets, are clustered at the state-industry level. ***: Significant at 1%, **: Significant at 5%, *: Significant at 10%.

**Table A7.** Economic reforms, labour market flexibility and employment in informal enterprises (1990–2001): Analysis based on the eight-firm concentration ratio (CR8) and the Herfindahl-Hirschman Index (HHI) in the formal sector in 1990.

| | Baseline (All Firms) | Firms in Industries with CR8 above Median in 1990 (Less Competitive Formal Sector) | Firms in Industries with CR8 below Median in 1990 (More Competitive Formal Sector) | Firms in Industries with HHI above Median in 1990 (Less Competitive Formal Sector) | Firms in Industries with HHI below Median in 1990 (More Competitive Formal Sector) |
|---|---|---|---|---|---|
| Final goods tariffs | −0.132 | −0.198 | −0.114 | −0.210 | −0.111 |
| | (0.113) | (0.175) | (0.074) | (0.194) | (0.074) |
| Final goods tariffs * FLEX 2 | 0.151 | 0.177 | 0.172 * | 0.185 | 0.160* |
| | (0.097) | (0.158) | (0.092) | (0.178) | (0.088) |
| Input tariffs | 0.576 * | 0.028 | 0.254 | 0.503 | 0.167 |
| | (0.327) | (0.447) | (0.378) | (0.482) | (0.416) |
| Input tariffs * FLEX 2 | −0.178 | −0.173 | −0.197 | −0.297 | −0.109 |
| | (0.168) | (0.276) | (0.182) | (0.294) | (0.186) |
| Delicensing | 0.108 *** | 0.077 ** | 0.093 ** | 0.074 ** | 0.106 ** |
| | (0.030) | (0.033) | (0.042) | (0.035) | (0.047) |
| Delicensing * FLEX 2 | −0.088 ** | −0.105 * | −0.008 | −0.108 ** | −0.022 |
| | (0.037) | (0.054) | (0.037) | (0.053) | (0.039) |
| FDI reform | 0.024 | −0.078 | 0.089 *** | −0.057 | 0.086 *** |
| | (0.028) | (0.071) | (0.033) | (0.067) | (0.033) |
| FDI reform * FLEX 2 | 0.076 * | 0.100 | 0.074 * | 0.104 | 0.076 * |
| | (0.041) | (0.111) | (0.041) | (0.110) | (0.041) |
| **Flexible labour markets: Effects of changes in final goods tariffs** | | | | | |
| **(Final goods tariffs)*(1 + FLEX 2)** | 0.019 | −0.021 | 0.058 | −0.025 | 0.050 |
| Standard Error | 0.078 | 0.135 | 0.070 | 0.149 | 0.070 |
| *p*-value (combined effect = 0) | 0.809 | 0.878 | 0.406 | 0.866 | 0.482 |
| **Flexible labour markets: Effects of changes in input tariffs** | | | | | |
| **(Input tariffs)*(1 + FLEX 2)** | 0.398 | −0.144 | 0.056 | 0.207 | 0.058 |
| Standard Error | 0.300 | 0.415 | 0.398 | 0.427 | 0.437 |
| *p*-value (combined effect = 0) | 0.185 | 0.728 | 0.887 | 0.629 | 0.895 |
| **Flexible labour markets: Effects of delicensing** | | | | | |
| **(Delicensing)*(1 + FLEX 2)** | 0.019 | −0.028 | 0.084 | −0.034 | 0.084 |
| Standard Error | 0.038 | 0.056 | 0.041 | 0.055 | 0.043 |
| *p*-value (combined effect = 0) | 0.613 | 0.616 | 0.038 | 0.529 | 0.049 |
| **Flexible labour markets: Effects of FDI reform** | | | | | |
| **(FDI reform)*(1 + FLEX 2)** | 0.099 | 0.022 | 0.163 | 0.047 | 0.162 |
| Standard Error | 0.043 | 0.114 | 0.049 | 0.112 | 0.050 |
| *p*-value (combined effect = 0) | 0.021 | 0.846 | 0.001 | 0.677 | 0.001 |
| State FE | Yes | Yes | Yes | Yes | Yes |
| Year FE | Yes | Yes | Yes | Yes | Yes |
| Industry FE | Yes | Yes | Yes | Yes | Yes |
| Observations | 195,789 | 87,156 | 108,633 | 93,680 | 102,109 |
| R-squared | 0.164 | 0.126 | 0.201 | 0.120 | 0.209 |

Dependent variable: natural logarithm of number of paid employees; 'FE' denotes fixed effects. Standard errors, in brackets, are clustered at the state-industry level. ***: Significant at 1%, **: Significant at 5%, *: Significant at 10%.

**Table A8.** Economic reforms and informal sector employment: Industry-level effects for enterprise numbers (1990–2001) based on the eight-firm concentration ratio (CR8) and the Herfindahl-Hirschman Index (HHI) in the formal sector in 1990.

| | Dependent Variable: ln (Number of Formal Firms in Three-Digit Industry) | | | | |
|---|---|---|---|---|---|
| | **All Industries** | **Industries with CR8 above Median in 1990 (Less Competitive Formal Sector)** | **Industries with CR8 below Median in 1990 (More Competitive Formal Sector)** | **Industries with HHI above Median in 1990 (Less Competitive Formal Sector)** | **Industries with HHI below Median in 1990 (More Competitive Formal Sector)** |
| **A: All states** | | | | | |
| Final goods tariffs | 0.202 | 0.143 | 0.234 | 0.206 | 0.173 |
| | (0.255) | (0.214) | (0.442) | (0.200) | (0.447) |
| Input tariffs | −3.083 ** | −2.754 * | −0.215 | −3.074 *** | 0.398 |
| | (1.475) | (1.439) | (1.774) | (0.879) | (1.875) |
| Delicensing | 0.184 | 0.320 *** | −0.136 | 0.381 *** | −0.216 |
| | (0.130) | (0.092) | (0.266) | (0.097) | (0.280) |
| FDI reform | 0.178 | 0.243 | −0.052 | 0.223 | −0.092 |
| | (0.137) | (0.174) | (0.220) | (0.172) | (0.238) |
| Observations | 378 | 156 | 222 | 159 | 219 |
| R-squared | 0.112 | 0.426 | 0.124 | 0.450 | 0.090 |
| **B: States with flexible labour markets (FLEX 2 = 1)** | | | | | |
| Final goods tariffs | 0.428 | 1.095 ** | −0.045 | 1.087 ** | −0.166 |
| | (0.416) | (0.541) | (0.469) | (0.496) | (0.511) |
| Input tariffs | −1.605 | −4.165 | −0.380 | −4.706 * | 1.772 |
| | (1.893) | (3.384) | (2.245) | (2.559) | (1.385) |
| Delicensing | −0.064 | 0.190 | −0.419 | 0.183 | −0.536 * |
| | (0.147) | (0.124) | (0.289) | (0.126) | (0.272) |
| FDI reform | 0.423 ** | 0.262 *** | 0.263 | 0.231 *** | 0.291 |
| | (0.169) | (0.084) | (0.308) | (0.069) | (0.310) |
| Observations | 327 | 132 | 195 | 138 | 189 |
| R-squared | 0.132 | 0.367 | 0.082 | 0.376 | 0.122 |
| **C: States with inflexible labour markets (FLEX 2 = 0)** | | | | | |
| Final goods tariffs | 0.237 | −0.121 | 0.592 | −0.035 | 0.544 |
| | (0.351) | (0.130) | (0.612) | (0.163) | (0.599) |
| Input tariffs | −3.946 * | −2.653 *** | −0.117 | −2.755 *** | −0.286 |
| | (2.038) | (0.638) | (2.483) | (0.423) | (3.005) |
| Delicensing | 0.312 ** | 0.335 *** | 0.153 | 0.449 *** | 0.103 |
| | (0.147) | (0.059) | (0.225) | (0.136) | (0.228) |
| FDI reform | 0.142 | 0.150 | −0.028 | 0.109 | −0.097 |
| | (0.129) | (0.094) | (0.201) | (0.083) | (0.217) |
| Observations | 357 | 141 | 216 | 144 | 213 |
| R-squared | 0.147 | 0.624 | 0.174 | 0.473 | 0.121 |

Dependent variable: natural logarithm of industry level number of formal enterprises. All regressions include a constant and industry and year fixed effects, and are weighted by pre-reform (1990) levels of the dependent variable. Standard errors, in brackets, are robust to heteroscedasticity. ***: Significant at 1%, **: Significant at 5%, *: Significant at 10%.

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
