# Peer review of "Economic Reform, Labour Markets and Informal Sector Employment: Evidence from India"

_economies, doi:10.3390/economies7020055_

Round 1
Reviewer 1 Report
The article is very interesting. I only suggest that you include appropriate references to the international literature.
Author Response
Thank you for your feedback - attached is my response to your comment(s).
Kind regards,
Nihar Shembavnekar

Reviewer 2 Report
General comment:This paper examined the extent to which differences in regional labour market flexibility shaped the impact of unanticipated economic reforms on employment in informal manufacturing companies in India.
Introduction:Focus more on the objective of the paper, the importance of your study and results.
The literature review should be extended. Add more references and place your research in the theoretical mainstream.
Methodology: Improvement in the section of methodology is required. Explain the limits and the advantages of the methods and provide more economic comments for introducing these methods.
Results:The results are reported without enough economic comments. Comparisons with previous studies are required. What are the economic implications of these results?
Discussion:Provide more economic comments for the results. The regression models should be validated (check errors homoskedasticity, normality, independence). Some coefficients in the regression models are not significant. Propose alternative methods.
Bibliography/References:More references are needed.
Other remarks:The language could be improved.

Author Response

(The authors gave the same response as above.)

Reviewer 3 Report
This paper examines to what extent the difference in regional labor market flexibility shapes the impact of unanticipated trade reforms on employment in informal manufacturing enterprises in India (1990-2001). Overall, it is a well-written manuscript so that I think it is suitable for publication after minor revision.
One (minor) concern in my mind is that it is too long. The abstract should be more concise (ideally less than 100-150 words?) Also, the whole manuscript can be reduced in 25 pages.
Author Response

(The authors gave the same response as above.)

Reviewer 4 Report
Comments to economies-443788
Overall, the topic is timely and will be of interest to the readers of the journal. However, the paper does not flow logically in writing a great relevance of the scientific inquires. Formally, the paper is well structured and ordered as follows:
Title
A title is not precise. A precise title should be “Economic Reform, Labour Markets, and Informal Sector: Evidence from India.”
Abstract
Abstract is clearly in which authors used what types of methods, sample and procedure, data collection tool and date, and data analysis.
The keyword is not precise. A precise keyword should be “economic reform; informal sector, labour market flexibility, India”
1. Introduction
The author did not justify well why the study is needed. The introduction doesn’t provide a clear picture of the research problem at hand.
1. The introduction does not highlight the importance of economic reform that impact of informal sector in the case of India?
2. How these labours have entered into the informal labour market?
3. What is sector plays important roles of informal employment?
4. Lacks one important evidence supported the quality of information such as number and sources (see line 36 through 39, page 1, for example, 99% of business and 80% has employed Indian labour)
5. Objectives of this study are not providing a clearer. However, the following sentences are not clear and do not make much sense, particularly, in the term on line 70 through 73, page 2.
6. In line 75 through 87, page 2. There is no evidence support, are public sector and private sector recorded about 10.8% in which increases employed informal labour, informal enterprise is arising equivalent to 9,9%, at the broader informal enterprise 32%, and FDI has 51%.
2. Context
Contexts lack of intensive review the core issues of sequential economic reform that the effect of the informal sector.
1. Review of each period in India economic reform policy has affected of informal employment activities, starting from five-years, according to the government policy records.
2. The characteristic of informal labour employed in the labour market (background, education, experiences, skills work, employment features, employment opportunity, and incomes).
3. Data
According to review the context of India economic reform between 1985 and 1997 but the author pilots the data only three periods. Figure 1 (b), starting from 1990, 1995, and 2001. Figure 2 (a, b), starting from 1989, but in (c) the author starting from 1989-2000.
4. Methods
Methods should be ordered of data, such as, sample and procedures, measures, empirical model, and data analysis would also be helpful.
5. Results
The author needs to be carefully detailed exactly present the results, which follows the data in section 3 will be affected by the substances in the method in sector 4. In line 501 (5.1. Baseline regression: Enterprise level) changed to 5.1. Ordinary Least Squares Result. In line 578 (5.2. Industry levels results) changed to 5.2. Robustness Results.
6. Conclusion
The paper lacks one important part which is clearly conclusion will be affected by the substantial changes suggested above.
6. Appendix
If the tables do not affect the results, which submits in the mode of Supplementary Data. If yes, substance to the study, moving it into the results.

Author Response

(The authors gave the same response as above.)

Round 2
Reviewer 2 Report
- Put text after Table 3. Do not end a section with a table.
- all the proposed models are not actually valid
-do you have Pooled OLS estimates in Table 4 and Table 6 or fixed-effects models? It seems that the author does not know exactly the difference between these models
-the information in Appendices is redundant since tables are provided in the content of the paper
Author Response
Thank you for your comments. I have attempted to address them - please find my responses attached.
Regards,
Nihar Shembavnekar

Reviewer 4 Report
I not convinced with the manuscript has been revised a point-by-point as follows all comments:
Point 1. Introduction
1. The introduction does not highlight the importance of economic reform that impact of informal sector in the case of India?
2. How these labours have entered into the informal labour market?
3. What is sector plays important roles of informal employment?
Point 2. Context
1. The characteristic of informal labour employed in the labour market (background, education, experiences, skills work, employment features, employment opportunity, and incomes).
Point 3. Conclusion
1. Lacks one important part which is clearly conclusion
Point 4. Appendix
1. Table A1 and Table A2. I’m not clear on what statistical is significantly, for example, -0.000000*** is significant at 1%
2. The decimal number not over than three numbers (.000)
Author Response

(The authors gave the same response as above.)
